

# Twisted holography of defect fusions

## Jihwan Oh[1*] and Yehao Zhou[2]

**1** Mathematical Institute, University of Oxford,
Woodstock Road, Oxford, OX2 6GG, United Kingdom
**2** Perimeter Institute for Theoretical Physics,
31 Caroline St. N., Waterloo, ON N2L 2Y5, Canada

* jihwan.oh@maths.ox.ac.uk

## Abstract

In the twisted M-theory setting, various types of fusion of M2 and M5 branes induce coproducts between the algebra of operators on M2 branes and the algebra of operators on M5 branes. By doing a perturbative computation in the gravity side, which is captured by the 5d topological holomorphic $U(1)$ Chern-Simons theory, we reproduce the non-perturbative coproducts.

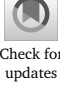

# 1   Introduction

Twisted holography [1–11][1] is a new and intriguing subject. Applying the familiar notion of a topological twist [12, 13] and an Omega background [19–22] of a supersymmetric quantum field theory on the dual supergravity, we can match the protected sub-sector of the field theory and the supergravity. One of the outstanding advantages of the Omega deformation of a twisted theory is that it gives a natural quantization of a ring of physical observables in the protected subsector, and endows an algebraic structure on the set of observables. Therefore, analyzing the algebraic structure of physical observables is a natural theme in the study of twisted holography.

We will be mainly concerned with the algebraic structure of the twisted M-theory [3, 4] in the presence of M2 branes and M5 branes. The algebra of observables on M2 and M5 branes is known to form a 1-shifted affine Yangian of $\widehat{gl}(1)$ [23–27][2], which we will call $\mathcal{A}$, and $\mathcal{W}_\infty$ algebra [28–30], respectively. Importantly, the algebras have three parameters $\epsilon_1$, $\epsilon_2$, $\epsilon_3$, which are the parameters of Omega deformations turned on three complex planes as a part of the eleven-dimension supergravity background. Depending on the orientations of the M2 branes(extending over one of the three complex planes) and the M5 branes(extending over two of the three complex planes) on the three Omega deformed planes, the description of the theories on the membrane worldvolume changes; however, both of $\mathcal{A}$ and $\mathcal{W}_\infty$ have triality [7, 31, 32] under the cyclic permutation of the deformation parameters.

At this point, one may wonder about the algebraic structure of a network of M2 and M5 branes extending over different complex planes. In [7], the authors conjectured a fusion of $\mathcal{A}$'s and interpreted an end of M2 branes on M5 branes as a degenerate module of a truncated version of $\mathcal{W}_\infty$ [31]. Moreover, recently the authors of [11] discovered a full algebraic structure governing intersecting M2-M5 branes. Key algebraic relations used to assemble the elements of the brane system are $\Delta_{\mathcal{A},\mathcal{A}}$, $\Delta_{\mathcal{A},\mathcal{W}_\infty}$, $\Delta_{\mathcal{W}_\infty,\mathcal{W}_\infty}$, coproducts of $\mathcal{A}$ and $\mathcal{W}_\infty$. They are induced by properly defined fusions of M2 and M5 branes. The strategy of [11] was to use a free field realization of both $\mathcal{A}$ and $\mathcal{W}_\infty$ algebras. This is the boundary field theory derivation in the context of the twisted M-theory.

The goal of this paper is to reproduce the coproducts of the M2-M5 brane system by doing a perturbative computation in the gravity side of the twisted M-theory. By the gravity side of the twisted M-theory, we mean the 5d topological holomorphic Chern-Simons theory, which is obtained as a result of localization of the Omega deformed twisted M-theory. The philosophy of our approach is simple to state. By probing the entire theory enriched with defects using the perturbative method[3], we will decode the non-perturbative algebraic structure of the defects. Moreover, we will prove that $\mathcal{A}$ equipped with $\Delta_{\mathcal{A},\mathcal{A}}$ satisfies the axioms of the vertex coalgebra.

---

[1]There are another lines of development in [14–18]. It would be nice to understand the relation between two sets of references.

[2]We thank Kevin Costello, who pointed out more math references.

[3]A similar set-up but using a non-perturbative method to find the algebraic data of a coupled system can be found in the bootstrap program for a BCFT, for instance [33].

We interpret the coproduct $\Delta_{\mathcal{A},\mathcal{A}} : \mathcal{A} \to \mathcal{A} \otimes \mathcal{A}$ as a fusion of two Wilson lines and the coproduct $\Delta_{\mathcal{W}_\infty,\mathcal{W}_\infty} : \mathcal{W}_\infty \to \mathcal{W}_\infty \otimes \mathcal{W}_\infty$ as a fusion of two surface defects, and compute the OPE of both defects in the 5d Chern-Simons theory background. Importantly, the quantum corrections in the coproduct relations are captured by 1-loop Feynman diagrams in the perturbation theory of the 5d Chern-Simons theory coupled with the defects.

Consistent with the logic under [11], which was used to explain the mixed coproduct $\Delta_{\mathcal{A},\mathcal{W}_\infty} : \mathcal{A} \to \mathcal{A} \otimes \mathcal{W}_\infty$, we will impose gauge invariance of the intersecting M2, M5 brane configuration coupled to the 5d Chern-Simons theory, and reproduce the mixed coproduct. Again, the quantum corrections in the $\mathcal{A} \to \mathcal{A} \otimes \mathcal{W}_\infty$ coproduct are captured by 1-loop Feynman diagrams in the 5d Chern-Simons theory that have vertices on both kinds of defects.

**Plan of our paper**

We will begin by reviewing some necessary background for our study in §2. Here, we define the twisted M-theory background and introduce M2 and M5 branes. Then, we briefly summarize the algebras of operators on the M2 and M5 branes in §2.1, §2.2 and discuss an embedding of $\mathcal{A}$ into $\mathcal{W}_\infty$ in §2.3. Lastly, we review the coproducts of $\mathcal{A}$ and $\mathcal{W}_\infty$ in §2.4.

We will discuss our main results in §3. We first present the result in §3.1 and §3.2. Using the ingredients of Feynman diagrams of the defect enriched 5d Chern-Simons theory collected in §3.3, we will give a holographic derivation for the coproduct $\Delta_{\mathcal{A},\mathcal{A}} : \mathcal{A} \to \mathcal{A} \otimes \mathcal{A}$ in §3.4, $\Delta_{\mathcal{A},\mathcal{W}_\infty} : \mathcal{A} \to \mathcal{A} \otimes \mathcal{W}_\infty$ in §3.5, and $\Delta_{\mathcal{W}_\infty,\mathcal{W}_\infty} : \mathcal{W}_\infty \to \mathcal{W}_\infty \otimes \mathcal{W}_\infty$ in §3.6. Moreover, we present our conjecture on the fusion of transverse surface defects in §3.7. The quantum corrections on the basic coproducts truncate at the first order. We prove this fact in §3.8 using the Feynman diagrams.

Finally, we conclude and list some open questions that we would like to revisit later in §4.

In Appendix A, we prove the vertex coalgebra structure of $\mathcal{A}$.

# 2 M2, M5 brane algebra and coproducts in the twisted M-theory

In this section, we will review various concepts in the twisted M-theory to set up a convention to use in §3, where we will present our main result. Most of the material in this section is already known and discussed in detail in many other references. We will try to collect all necessary backgrounds here for completeness; however, for more details we direct the reader to [3, 4, 7, 9, 11].

By the $\Omega$-deformed M-theory, we refer to 11-dimensional supergravity, which is topological in 7 directions and holomorphic in 4 directions. The twisted background is specified by a triple $(g, \Psi, C)$, which consists of a metric, a bosonic ghost taking some nonzero value, and the M-theory 3-form field. $\Psi$ that satisfies $\Psi^2 = 0$ can be thought of as a local supersymmetry transformation parameter $\epsilon(x^\mu)$ that parametrizes local supersymmetry(or supergravity).

When we study branes in such a background, we turn off the spacetime dependence of $\Psi$ on $x^\mu$, it simply reduces to the global supersymmetry transformation parameter that identifies a supercharge whose cohomology defines a topological(possibly holomorphic) quantum field theory on the brane.

The twisted supergravity background is compatible with an $\Omega$ background [3]. The deformed background with a deformation parameter $\epsilon_i$ is specified by the triple $(g, \Psi_\epsilon, C)$ again, but with a modification on $\Psi_\epsilon$ such that $\Psi_\epsilon^2 = \mathcal{L}_{V_\epsilon}$, where $\mathcal{L}_{V_\epsilon}$ is a rotation generator acting on a plane, where Omega background $\Omega_\epsilon$ is turned on. We may turn on $\Omega_\epsilon$ on multiple planes. In the background we are interested in, we have, out of the 7 topological directions, 6 directions equipped with an Omega background $\Omega_{\epsilon_1} \times \Omega_{\epsilon_2} \times \Omega_{\epsilon_3}$ with a Calabi-Yau condition

$\epsilon_1 + \epsilon_2 + \epsilon_3 = 0.$

$$\text{11d background: } (\mathbb{R}_t \times \mathbb{C}_{\epsilon_1} \times \mathbb{C}_{\epsilon_2} \times \mathbb{C}_{\epsilon_3})_{\text{topological}} \times (\mathbb{C}_z \times \mathbb{C}_w)_{\text{holomorphic}}. \tag{1}$$

M-theory on this background localizes on 5d $U(1)$ Chern-Simons theory [3, 34](also, see the nice description of the related 4d Chern-Simons theory [35] in [4]) with a leading order action given as [4]

$$\frac{1}{\sigma_3} \int_{\mathbb{R}_t \times \mathbb{C}_z \times \mathbb{C}_w} (AdA + A\{A,A\})dzdw, \tag{3}$$

where $\sigma_3^{-1} = (\epsilon_1 \epsilon_2 \epsilon_3)^{-1}$ is the equivariant volume of $\mathbb{C}_{\epsilon_1} \times \mathbb{C}_{\epsilon_2} \times \mathbb{C}_{\epsilon_3}$ and

$$A = A_t dt + A_{\bar{z}} d\bar{z} + A_{\bar{w}} d\bar{w}, \tag{4}$$

and $\{A,A\}$ is the holomorphic Poisson bracket defined as

$$\{A,A\} = \frac{\partial A}{\partial z}\frac{\partial A}{\partial w} - \frac{\partial A}{\partial w}\frac{\partial A}{\partial z}. \tag{5}$$

This is nonzero, since $A$ is a 1-form, not a function.

In this background, we may introduce $N$ M2 branes and $N'$ M5 branes. M2 and M5 branes extend in 1 and 2 real directions, respectively in the 5d Chern-Simons theory, and can be considered as line and surface defects with their degrees of freedom interacting with the 5d Chern-Simons theory. We will review the coupling between those defects and the 5d Chern-Simons theory in §2.1, §2.2.

Table 1: M2, M5-brane and 5d Chern-Simons theory. In general, $M2$ branes may extend over $\mathbb{R}_t \times \mathbb{C}_{\epsilon_i}$ and $M5$ branes may extend over $\mathbb{C}_{z \text{ or } w} \times \mathbb{C}_{\epsilon_i} \times \mathbb{C}_{\epsilon_j}$, where $i, j \in \{1, 2, 3\}$.

| | 0 | 1 | 2 | 3 | 4 | 5 | 6 | 7 | 8 | 9 | 10 |
|---|---|---|---|---|---|---|---|---|---|---|---|
| Geometry | $\mathbb{R}_t$ | $\mathbb{C}_{\epsilon_1}$ | | $\mathbb{C}_z^2$ | | $\mathbb{C}_w^2$ | | $\mathbb{C}_{\epsilon_3}$ | | $\mathbb{C}_{\epsilon_2}$ | |
| $M2$ | × | × | × | | | | | | | | |
| $M5$ | | | | | × | | × | × | × | × | × |
| 5d CS | × | | | × | × | × | × | | | | |

Taking either the large $N$ or $N'$ limit[5], we can discuss a holographic duality [36–38] between the 5d Chern-Simons theory and the topologically twisted worldvolume theory of membranes, which is further deformed by the Omega background.

Twisted holography has a nice feature that we can compare the operator algebras on each side of the duality and construct an isomorphism between two operator algebras. Moreover, demanding gauge-invariant couplings between the 5d Chern-Simons theory and the worldvolume theory of each membrane, we can derive the operator algebra of the field theory on the membrane by analyzing a certain collection of Feynman diagrams [4, 9]. In other words, the

---

[4]In the original paper of Costello [3], the action took a different form as

$$\frac{1}{\epsilon_1} \int_{\mathbb{R}_t \times \mathbb{C}_z \times \mathbb{C}_w} (AdA + A *_{\epsilon_2} A *_{\epsilon_2} A)dzdw, \tag{2}$$

where $*$ is a Moyal product combined with the wedge product. The equivalent action (3), which makes the triality among $\epsilon_i$'s manifest, was suggested in [7], and it will be more convenient in our computation.

[5]Taking both $N, N'$ to be large simultaneously is not required for this duality, but one can take either $N$ or $N'$ large. For instance, if we take large $N$ limit, we consider $N'$ M5 branes as defects in the twisted M2 brane holography.

OPE on the membrane receives corrections from coupling to the bulk Chern-Simons theory and the corrections are computed by Feynman diagrams. A caveat is that we only compute the OPE at the first order of $\hbar$; however, a theorem of Costello [4, Theorem 16.0.1] indicates that OPE in higher order of $\hbar$ is fully determined by the algebra of classical operators (i.e. undeformed operator algebra) and the first order deformation in $\hbar$.

The holographic duality of the twisted M-theory is interpreted as a Koszul duality in the original reference [4]. The algebra of classical local observables of 5d $\mathfrak{gl}(k)$ Chern-Simons theory $\mathrm{Obs}^{\mathrm{cl}}(\mathrm{CS})$ is the Chevalley-Eilenberg complex of the Lie algebra $\mathfrak{gl}(k) \otimes \mathrm{Diff}_{\epsilon_2}(\mathbb{C})$ [4]. The Koszul dual algebra of $\mathrm{Obs}^{\mathrm{cl}}(\mathrm{CS})$ is the universal enveloping algebra of $\mathfrak{gl}(k) \otimes \mathrm{Diff}_{\epsilon_2}(\mathbb{C})$. Moreover, we can turn on a quantum deformation parameter $\epsilon_1$ and the Koszul duality holds after the deformation, i.e. $\mathrm{U}_{\epsilon_1}(\mathfrak{gl}(k) \otimes \mathrm{Diff}_{\epsilon_2}(\mathbb{C}))$ is the Koszul dual of $\mathrm{Obs}^{\mathrm{q}}(\mathrm{CS}) \cong \mathrm{C}^*_{\epsilon_1}(\mathfrak{gl}(k) \otimes \mathrm{Diff}_{\epsilon_2}(\mathbb{C}))$. On the other hand, it is shown in [4] that there is a surjective map

$$\mathrm{U}_{\epsilon_1}(\mathfrak{gl}(k) \otimes \mathrm{Diff}_{\epsilon_2}(\mathbb{C})) \to \mathcal{A}^{(N)}, \tag{6}$$

where $\mathcal{A}^{(N)}$ is the M2 brane algebra discussed in the next subsection. This map is compatible with the surjective map $\mathcal{A}^{(N+1)} \to \mathcal{A}^{(N)}$. Moreover, the intersection of the kernels of $\mathrm{U}_{\epsilon_1}(\mathfrak{gl}(k) \otimes \mathrm{Diff}_{\epsilon_2}(\mathbb{C})) \to \mathcal{A}^{(N)}$ for all $N$ is zero. In this sense, we call $\mathrm{U}_{\epsilon_1}(\mathfrak{gl}(k) \otimes \mathrm{Diff}_{\epsilon_2}(\mathbb{C}))$ the *large N limit* of $\mathcal{A}^{(N)}$, and denote it by $\mathcal{A}$. In summary, the large N limit of the M2 brane algebra is the Koszul dual of the 5d Chern-Simons algebra. From now on, we will sometimes use the term Koszul dual to refer to the twisted holographic dual.

After we understand the worldvolume algebras $\mathcal{A}$ of M2 branes and $\mathcal{W}_\infty$ of M5 branes from §2.1, §2.2, the next step is to study fusions between the algebras $\mathcal{A}$ and $\mathcal{W}_\infty$ [11]. To do that, it is helpful to understand the relation between $\mathcal{A}$ and $\mathcal{W}_\infty$, as we are mixing them while performing the fusion. We will briefly go over the relation in §2.3 and review the coproduct in §2.4.

## 2.1 M2 brane algebra $\mathcal{A}$

One of the UV descriptions of the M2 branes worldvolume theory is 3d $\mathcal{N} = 4$ U(N) gauge theory with 1 adjoint hypermultiplet(with scalars $X, Y$) and $K$ fundamental hypermultiplets(with scalars $I, J$). Under the topological twist applied on the M2 brane worldvolume, given by the twisted supergravity background, we may restrict our attention to the Higgs branch chiral ring that fully captures the operators in the Q-cohomology. This consists of gauge-invariant operators made of $X, Y, I, J$, which are further divided by the ideal generated by the F-term relation,

$$X^a_c Y^c_b - X^c_b Y^a_c + I_b J^a = \epsilon_2 \delta^a_b. \tag{7}$$

An $\Omega_{\epsilon_1}$ background deforms the chiral ring into an algebra $\mathcal{A}$ [39, 40] by making Poisson brackets into commutators

$$\left[X^a_b, Y^c_d\right] = \epsilon_1 \delta^a_d \delta^c_b, \quad [J^b, I_a] = \epsilon_1 \delta^b_a, \tag{8}$$

and the theory localizes on one-dimensional TQM(topological quantum mechanics)

$$\frac{1}{\epsilon_1} \int_{\mathbb{R}_t} \mathrm{Tr}[\epsilon_2 A_t + X D_t Y + J D_t I] dt. \tag{9}$$

Note that $\epsilon_1$ is a deformation parameter, and $\epsilon_2$ is an FI parameter of the 3d $\mathcal{N} = 4$ gauge theory. This 1d TQM was originally studied in [41] using the technique developed in [42](see also [43,44]); a similar discussion using the Coulomb branch algebra [45] can be found in [46].

The algebra $\mathcal{A}^{(N)}$ is generated by

$$t_{m,n} = \frac{1}{\epsilon_1} \text{STr} X^m Y^n, \tag{10}$$

where $STr[\bullet]$ means to take a trace of a symmetrization of a polynomial $\bullet$. One may wonder about the $N$ dependence of the algebra. We can find it in $t_{0,0}$, since the trace of $N \times N$ unit matrix is $N$. As we take a large $N$ limit, it becomes an element of the algebra $U_{\epsilon_1}(\mathfrak{gl}(1) \otimes \text{Diff}_{\epsilon_2}(\mathbb{C}))$, not a number. And we will denote the large N limit by $\mathcal{A}$.

A seed set of commutation relations was proposed in [10] and was proved in [9].

$$
\begin{aligned}
&\big[t_{0,0}, t_{c,d}\big] = 0, \\
&\big[t_{1,0}, t_{c,d}\big] = d t_{c,d-1}, \quad \big[t_{0,1}, t_{c,d}\big] = -c t_{c-1,d}, \\
&\big[t_{2,0}, t_{c,d}\big] = 2d t_{c+1,d-1}, \quad \big[t_{1,1}, t_{c,d}\big] = (d-c) t_{c,d}, \quad \big[t_{0,2}, t_{c,d}\big] = -2c t_{c-1,d+1}, \\
&\big[t_{3,0}, t_{c,d}\big] = 3d t_{c+2,d-1} + \sigma_2 \frac{d(d-1)(d-2)}{4} t_{c,d-3} \\
&\quad + \frac{3}{2}\sigma_3 \sum_{m=0}^{d-3} \sum_{n=0}^{c} \frac{\binom{m+n+1}{n+1}(n+1)\binom{d-m+c-n-2}{c-n+1}(c-n+1)}{\binom{d+c}{c}} t_{n,m} t_{c-n,d-3-m},
\end{aligned} \tag{11}
$$

where

$$\sigma_2 = \epsilon_1^2 + \epsilon_1 \epsilon_2 + \epsilon_2^2, \quad \sigma_3 = \epsilon_1 \epsilon_2 \epsilon_3. \tag{12}$$

The parameters $\sigma_2$, $\sigma_3$ indicates that $\epsilon_1, \epsilon_2, \epsilon_3$ appear symmetrically in the algebra; however, this is not obvious from the formulation of the algebra (7),(8). One intuitive way to understand the triality structure is to appeal to the self-mirror property of the 3d $\mathcal{N}=4$ theory. By the 3d mirror symmetry, Higgs branch of one theory is isomorphic to Coulomb branch of the mirror pair. As the theory is self-mirror, one can investigate the Coulomb branch algebra to get information about $\mathcal{A}$. The Coulomb branch algebra of the 3d $\mathcal{N}=4$ theory is known to be 1-shifted affine $gl(1)$ Yangian [24] and it inherits the triality property of affine $gl(1)$ Yangian [23]. On the other hand, one can directly compute the commutator of the Higgs branch algebra and show the above is correct [9].

With the seed relations shown in (11), one can recursively find all other commutation relations for a general pair of elements in $\mathcal{A}$ [10].

Note that $t_{2,0}, t_{1,1}, t_{0,2}$ generate a copy of $\mathfrak{sl}_2$ inside $\mathcal{A}$, and $\mathcal{A}$ itself is a direct sum of finite dimensional representations of $\mathfrak{sl}_2$: by the third line of (11), subspace spanned by $t_{a,b}, a+b = n$ is a representation space of spin $n/2$. This implies that the action of $\mathfrak{sl}_2$ on $\mathcal{A}$ integrates to an action of $\text{SL}_2(\mathbb{C})$ on $\mathcal{A}$. Physically, $\text{SL}_2(\mathbb{C})$ is the group of the linear automorphisms of $\mathbb{C}_z \times \mathbb{C}_w$ which preserves the commutator $[z,w] = \epsilon_2$, so it is a symmetry of the 5d Chern-Simons theory with M2 brane insertion at $z = w = 0$.

We have discussed so far the algebra of operators on the M2 branes on $\mathbb{R}_t \times \mathbb{C}_{\epsilon_1}$. One can orient the M2 branes in $\mathbb{C}_{\epsilon_2}$, $\mathbb{C}_{\epsilon_3}$ directions and get different algebras, which are obtained by cyclically permuting $\epsilon_1, \epsilon_2, \epsilon_3$ in $\mathcal{A}$. Doing so, one can easily see the commutation relations (11) do not change, but only the elements of the algebra (10) and the F-term relation (7) change.

We may understand the algebra $\mathcal{A}$ geometrically as transverse fluctuations of M2 brane in $\mathbb{C}_z \times \mathbb{C}_w$ direction in the 5d Chern-Simons spacetime. Formally, we can encode the fluctuations in a nice algebra: $U_{\epsilon_1}(gl_1 \otimes \mathbb{C}_{\epsilon_2}^2[z,w])$ a deformation of the universal enveloping algebra of the algebra of holomorphic functions on the non-commutative complex plane $\mathbb{C}_{\epsilon_2}^2$. Here, $\epsilon_1$ is the deformation parameter related to the Omega background $\Omega_{\epsilon_1}$ and $\epsilon_2$ is a non-commutativity parameter, which induces a commutator $[z,w] = \epsilon_2$. Although we suggested this intuitive

identification, using the heuristic description of the fluctuations of the M2 branes, the proof is nontrivial and is one of the main results of [4].

$U_{\epsilon_1}(gl_1 \otimes \mathbb{C}_{\epsilon_2}[z, w])$ is the gauge symmetry algebra preserving the trivial field configuration of the 5d Chern-Simons theory on $\mathbb{R}_t \times (\mathbb{C}_z \times \mathbb{C}_w \backslash \{0\})$(this is the back-reacted geometry). In other words, it is the algebra of operators in the 5d Chern-Simons theory. We can identify the classical coupling between $\partial_z^m \partial_w^n A$(the modes of 5d Chern-Simons theory) and $t_{m,n}$(M2 brane algebra elements) as

$$\int_{\mathbb{R}_t} t_{m,n} \partial_z^m \partial_w^n A. \tag{13}$$

Quantum mechanically, for the 5d Chern-Simons theory to be compatible with the M2 brane line defect, all correlation functions or Feynman diagrams that involve vertices on both the defect and the bulk should be invariant under the BRST transformation

$$Q_{\text{BRST}} A = dc + [A, c], \quad Q_{\text{BRST}} c = -\frac{1}{2}[c, c], \tag{14}$$

where $c$ is a scalar ghost. The bracket does not vanish in general even that we are considering U(1) gauge theory, since $\mathbb{C}_{\epsilon_2}^2$ is non-commutative.

## 2.2   M5 brane algebra $\mathcal{W}_\infty$

Let us fix the orientation of the $N'$ M5 branes so that they extend over $\mathbb{C}_w \times \mathbb{C}_{\epsilon_2} \times \mathbb{C}_{\epsilon_3}$. We are interested in the M5 brane theory on $\mathbb{C}_w$, as the M5 branes intersect with the 5d Chern-Simons theory along $\mathbb{C}_w$. For this, it is rather convenient to go to the IIa frame(by compactifying the M-theory circle $S^1 \in \mathbb{C}_{\epsilon_2}$). In the type IIa frame, the theory on $\mathbb{C}_w$ consists of D4-D6 strings, with 8 ND directions; this gives rise to a pair of chiral fermions $\psi$, $\psi'$ with a Lagrangian

$$\int_{\mathbb{C}_w} dz \, \text{Tr} \, \psi(\bar{\partial} + A)\psi'. \tag{15}$$

Also, the resulting algebra consists of modes of various currents labeled by its conformal dimension $n$: $W^{(n)} = \psi \partial_w^{n-1} \psi'$, where $n$ runs from 1 to $N'$. [3] proposed a mathematically rigorous way to take the large $N'$ limit and showed that the M5 brane algebra is $\mathcal{W}_\infty$.

Note that another intuitive way to understand the M5 brane algebra is via AGT set-up [47]. N' M5 brane worldvolume theory is the 6d (2,0) theory of $A_{N'-1}$ type on 1 holomorphic direction $\mathbb{C}_w$ and 4 topological directions $\mathbb{C}_{\epsilon_2} \times \mathbb{C}_{\epsilon_3}$ with an Omega background turned on both of topological planes [48]. Localizing on the locus of the Omega background, we get a $\mathcal{W}_\infty$ algebra on the holomorphic plane [48–51].

The coupling between the currents in the theory of the M5 branes and the gauge field of the 5d Chern-Simons theory is given by [6]

$$\int_{\mathbb{C}_w} dw \, W^{(m)} \partial_w^{m-1} A. \tag{16}$$

To see an explicit coupling between the m-th mode of $W_m^{(n)}$ and 5d gauge field, let us expand $W^{(n)}$ in $w$:

$$\sum_{m \in \mathbb{Z}} W_n^{(m)} \int_{\mathbb{C}} w^{-m-n} \partial_w^{m-1} A dw. \tag{17}$$

Therefore, the $n$-th mode of $W^{(m)}$ current $W_n^{(m)}$ couples to $w^{-m-n}(\partial_w^{m-1} A) dw$.

---

[6]A similar example of the surface operator was discussed in [56] in the context of 4d Chern-Simons theory.

Quantum mechanically, for the 5d Chern-Simons theory to be compatible with the surface defect from the M5 branes, all correlation functions or Feynman diagrams that involve vertices both to the defect and the bulk should be invariant under the BRST transformation $A \to dc + [c, A]$, where $c$ is a scalar ghost.

## 2.3 An embedding of $\mathcal{A}$ in $\mathcal{W}_\infty$

As one of our goal is to understand the fusion of two algebras $\mathcal{A}$ and $\mathcal{W}_\infty$ holographically, it is important to know the relation between two algebras. [11] showed that there is an embedding map $\rho$ from $\mathcal{A}$ to $\mathcal{W}_\infty$. $\mathcal{A}$ maps to a deformation of modes of $\mathcal{W}_\infty$ that annihilates the vacuum of the chiral algebra.

Physically, the embedding relation is not immediately clear, since $\mathcal{A}$ and $\mathcal{W}$ are associated to the algebra of operators on a topological line and the algebra of operators on a transverse holomorphic plane. Algebraically, we can understand the relation better by appealing to the relation of $\mathcal{A}$ and $\mathcal{W}_\infty$ to affine $gl(1)$ Yangian $\mathcal{Y}$. In §2.1, we explained that $\mathcal{A}$ is 1-shifted affine $gl(1)$ Yangian. It is also known that $\mathcal{W}_\infty$ is isomorphic to $\mathcal{Y}$ [28], and 1-shifted affine $gl(1)$ Yangian is a subalgebra of $\mathcal{Y}$ [23]. Hence, we can expect that there is an embedding map from $\mathcal{A}$ to $\mathcal{W}_\infty$.[7]

For our purpose, we only present the map for the first few elements of $\mathcal{A}$.

$$\rho(t_{0,n}) = W_n^{(1)},$$
$$\rho(t_{2,0}) = V_{-2} + \sigma_3 \sum_{n=1}^\infty n W_{-n-1}^{(1)} W_{n-1}^{(1)}, \tag{18}$$

where

$$V = W^{(3)} + \frac{2}{\psi_0} : W^{(1)} W^{(2)} : - \frac{2}{3} \frac{1}{\psi_0^2} : W^{(1)} W^{(1)} W^{(1)} :, \tag{19}$$

and $\psi_0$ is a central element of $\mathcal{W}_\infty$ algebra. (18) will be used in §3.5, where we derive a basic coproduct $\mathcal{A} \to \mathcal{A} \otimes \mathcal{W}_\infty$ using the 5d Chern-Simons theory. For details of the embedding, see Appendix A.7 of [11].

Let us make a remark on $\rho$ and its relationship with the defect couplings before ending this subsection. Superficially, there is a sharp conflict between the embedding map $\rho$ and two defect couplings (13), (17). Let us consider the LHS of the first equation of (18). According to (13), $t_{0,n}$ couples to $\partial_w^n A$; however, the RHS of the same equation $W_n^{(1)}$ couples to $w^{-n-1} A$. Also, considering the second line of (18), although $t_{2,0}$ couples to $\partial_z^2 A$, the leading term of $V_{-2}$, $W_{-2}^{(3)}$, couples to $w^{-1} \partial_w^2 A$.

This indicates the embedding map $\rho : \mathcal{A} \to \mathcal{W}_\infty$ induces a non-trivial "Koszul dual" morphism $^!\rho$ that maps a Koszul dual 5d gauge mode into another. Applying it to the examples described above, we get $^!\rho(\partial_w^n A) = w^{-n-1} A$ and $^!\rho(\partial_z^2 A) = w^{-1} \partial_w^2 A$. We have not attempted to figure out the induced morphism $^!\rho$, but we assumed that there exists such a map when we tried to match the tree-level Feynman diagrams in §3. It would be interesting to construct $^!\rho$ precisely.

## 2.4 Coproducts of M2, M5 brane algebra

Recently, [11] proposed a recipe to fuse $\mathcal{A}$ and $\mathcal{W}_\infty$. There are two types of fusion, which we will respectively call homogeneous fusion and heterotic fusion.

**The homogeneous fusion**

---

[7]We thank Davide Gaiotto for the illuminating discussion on the idea described in this paragraph.

Fusion :

$\mathcal{W}_1$ ——

—— $\mathcal{W}$ ——

$\mathcal{W}_2$ ——

$\Rightarrow \Delta_{\mathcal{A},\mathcal{A}} : \mathcal{A} \to \mathcal{A} \otimes \mathcal{A}$

Fusion :

$\mathcal{S}_1$

—— $\mathcal{S}$

$\mathcal{S}_2$

$\Rightarrow \Delta_{\mathcal{W}_\infty,\mathcal{W}_\infty} : \mathcal{W}_\infty \to \mathcal{W}_\infty \otimes \mathcal{W}_\infty$

Figure 1: The top figure schematically describes that the Wilson line fusion induces the coproduct in $\mathcal{A}$. The bottom figure shows the surface operator fusion induces the coproduct in $\mathcal{W}_\infty$.

The homogeneous fusion is between the same type of defects. As there are two types of defects, we have two homogeneous fusions: a fusion of line defects with each other and a fusion of surface defects with each other. The operation of the two homegenous fusions is given as follows

- Place two M2 branes at separate points in one of the holomorphic directions $\mathbb{C}_w$ and bring them together.

- Place two M5 branes at separate points in the topological direction $\mathbb{R}_t$ and bring them together.

We may consider this operation as an OPE of two defects $\mathcal{D}_1$, $\mathcal{D}_2$ that leads to a single defect $\mathcal{D}$. Therefore, we may ponder about the relation among the operator algebras $\mathcal{A}(\mathcal{D}_1)$, $\mathcal{A}(\mathcal{D}_2)$, $\mathcal{A}(\mathcal{D})$, associated to $\mathcal{D}_1, \mathcal{D}_2, \mathcal{D}$. The fusion process is a 2-to-1 operation from the bulk algebra point of view, and Koszul-dually [8] it induces an 1-to-2 operation, which will be called coproducts $\Delta_{\mathcal{A},\mathcal{A}}$, $\Delta_{\mathcal{W}_\infty,\mathcal{W}_\infty}$ on each $\mathcal{A}$ and $\mathcal{W}_\infty$.

$$\Delta_{\mathcal{A},\mathcal{A}} : \mathcal{A} \to \mathcal{A}_1 \otimes \mathcal{A}_2,$$
$$\Delta_{\mathcal{W}_\infty,\mathcal{W}_\infty} : \mathcal{W}_\infty \to \mathcal{W}_{\infty,1} \otimes \mathcal{W}_{\infty,2}. \tag{20}$$

Physically, we may see the existence of the coproducts in the bulk side through Feynman diagrams with a bulk 3-point vertex, which has two internal legs connecting to 2 defects participating in the fusion and 1 external leg.

We visualized the process so far in Figure 1.

Now, let us write down the representative example of the coproduct $\Delta_{\mathcal{A},\mathcal{A}}$ [11] that we will try to reproduce in the next section:

$$t_{2,0} \to t'_{2,0} + \tilde{t}_{2,0} + 2\sigma_3 \sum_{m,n \geq 0} \mathrm{d}_{m,n} t'_{0,m} \tilde{t}_{0,n} \tilde{w}^{-m-n-2}. \tag{21}$$

---

[8]Schematically, the Koszul dual algebra $^!A$ of an algebra $A$ has the functorial property that $\mathrm{Hom}_{\mathrm{algebra}}(^!A, B) \cong \text{Maurer-Cartan}(B \otimes A)$, where the Maurer-Cartan elements in $B \otimes A$ is interpreted as the coupling between two systems with algebra of local observables $B$ and $A$. Fusion of two line operators with operator algebra $^!A$ gives rise to a Maurer-Cartan element in $^!A \otimes {}^!A \otimes A$ and this induces a map $^!A \to {}^!A \otimes {}^!A$.

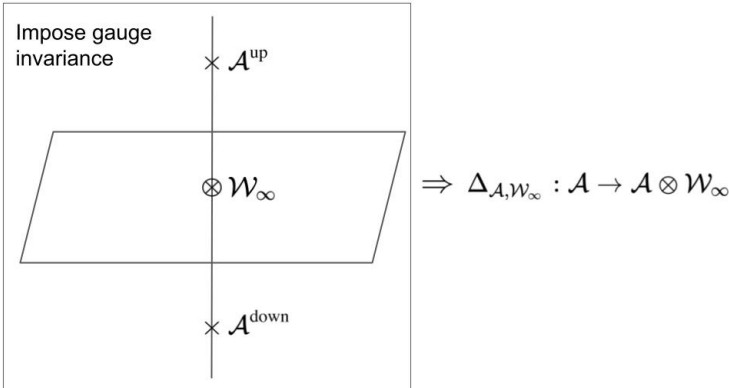

Figure 2: Imposing the gauge-invariance of the coupled system of the line defect and the surface defect induces the coproduct $\Delta_{\mathcal{A},\mathcal{W}_\infty}$.

$t_{2,0}$, $t'_{2,0}$, $\tilde{t}_{2,0}$ are elements of $\mathcal{A}$, $\mathcal{A}_1$, $\mathcal{A}_2$. $d_{m,n}$ is a combinatorial factor that depends on $m$ and $n$. $\tilde{w}$ is a separation of two line defects in the $\mathbb{C}_w$-plane.

The coproduct $\Delta_{\mathcal{A},\mathcal{A}}$ comes from the fusion of two Wilson lines. If we bring three Wilson lines together in the $\mathbb{C}_w$ plane, then they fuse without ambiguity, which means that the fusion is associative. Koszul-dually, this means that the coproduct $\Delta_{\mathcal{A},\mathcal{A}}$ should satisfy coassociativity in some sense, and this is mathematically captured by the notion of vertex coalgebra [62]. In Appendix A, we make our observation rigorous by proving that $\mathcal{A}$ equipped with $\Delta_{\mathcal{A},\mathcal{A}}$ satisfies the axioms of the vertex coalgebra.

**The heterotic fusion**

The heterotic fusion is between different types of defects: a line and a surface, or M2 branes intersecting with M5 branes. Different from the case of homogeneous fusions, which have a simple interpretation as an OPE of defects, the heterotic fusion is subtle. The coproduct for the heterotic fusion is induced by imposing a gauge-invariant condition on the M2-M5 junction configuration. Imposing gauge-invariance of the entire coupled system leads to the following schematic relation between various operators in the system:

$$t_{n,m}^{\mathrm{up}} \cdot O + W_{m-n}^{(n+1)} \cdot O + (\ldots) \cdot O - O \cdot t_{n,m}^{\mathrm{down}} = 0, \tag{22}$$

where $\mathcal{O}$ represents the junction between the line and the surface, and $(\ldots)$ is a sum of polynomials of elements of $\mathcal{A}$ and $\mathcal{W}_\infty$ that can be seen as quantum corrections. By arranging the terms in (22) as

$$O \cdot (t_{n,m}^{\mathrm{down}}) = (t_{n,m}^{\mathrm{up}} + W_{m-n}^{(n+1)} + (\ldots)) \cdot O, \tag{23}$$

and comparing the LHS and the RHS, we can notice that the gauge invariance induces a map between $\mathcal{A}$ and $\mathcal{A} \otimes \mathcal{W}_\infty$:

$$\Delta_{\mathcal{A},\mathcal{W}_\infty} : \mathcal{A} \to \mathcal{A} \otimes \mathcal{W}_\infty. \tag{24}$$

The representative example [11] of the coproduct $\Delta_{\mathcal{A},\mathcal{W}_\infty}$ is

$$t_{2,0} \to t_{2,0} + V_{-2} + \sigma_3 \sum_{n=1}^{\infty} n W_{-n-1}^{(1)} W_{n-1}^{(1)} + \sigma_3 \sum_{n=1}^{\infty} n W_{-n-1}^{(1)} t_{0,n-1}. \tag{25}$$

In the RHS, $t_{2,0}$ and $V_{-2}$ are implicitly $t_{2,0} \otimes 1$ and $1 \otimes V_{-2}$, so both are elements of $\mathcal{A} \otimes \mathcal{W}_\infty$.

# 3 Holographic derivation of the coproducts

In this section, we will give a twisted holographic derivation of the various coproducts, which we reviewed in the previous section.

The original derivation [11] of the coproducts $\Delta_{\mathcal{A},\mathcal{A}} : \mathcal{A} \to \mathcal{A} \otimes \mathcal{A}$ and $\Delta_{\mathcal{W}_\infty,\mathcal{W}_\infty} : \mathcal{W}_\infty \to \mathcal{W}_\infty \otimes \mathcal{W}_\infty$, which are induced by the homogeneous fusion, was purely algebraic, appealing to the free field realization of $\mathcal{A}$ and $\mathcal{W}_\infty$ [52,53]. We will explain how to take an OPE of two identical type defects and produce a single defect by computing 1-loop Feynman diagrams. The RHS of the coproducts $\Delta_{\mathcal{A},\mathcal{A}}$, $\Delta_{\mathcal{W}_\infty,\mathcal{W}_\infty}$ naturally emerges as a fusion coefficient of the resulting single defect. We will first state the result in §3.1 with a diagrammatic explanation. Using the various ingredients of the Feynman diagram collected in §3.3, we give an explicit Feynman diagram computation in §3.4, §3.6.

The philosophy of the argument that leads to the coproducts $\mathcal{A} \to \mathcal{A} \otimes \mathcal{W}_\infty$ was to impose the gauge-invariance of the intersecting M2-M5 configuration. [11] derived the coproduct by utilizing purely algebraic properties of $\mathcal{A}$ and $\mathcal{W}_\infty$. As the system couples to the bulk 5d Chern-Simons theory, imposing the gauge invariance implicitly assumes the gauge-invariance of the entire system. We will explain how to compute the possible gauge anomaly of a collection of Feynman diagrams, where defects interact with the bulk. By imposing the vanishing anomaly condition, we reproduce the coproduct $\mathcal{A} \to \mathcal{A} \otimes \mathcal{W}_\infty$. We will first state the result in §3.2 with a diagrammatic explanation and give an explicit Feynman diagram computation in §3.5.

In §3.7, we propose a conjecture about the fusion between two transverse surface defects. Different from the fusion between two parallel surface defects, we will see a line operator as one of the byproducts.

Note that the coproducts that we are dealing with are all truncated in the first order of $\sigma_3$. We prove the dual statement in the 5d Chern-Simons side in §3.8.

Our calculation is based on the integral technique developed in [54] in the context of 4d Chern-Simons theory. The authors discussed an OPE between two Wilson lines and show that it gives a composite Wilson line. We will sometimes rely on our previous paper [9], as well.

## 3.1 Holographic interpretation of the homogeneous fusion

Given two parallel Wilson lines, placed on the $\mathbb{C}_w$ plane at $w = 0$, $w = \tilde{w}$, when they approach each other, $\tilde{w} \to 0$, we obtain a single Wilson line. We will directly compute the OPE of two Wilson lines in the 5d Chern-Simons background using Feynman diagrams.

At the tree level, the OPE of two Wilson lines associated with $t'_{2,0}$, $\tilde{t}_{2,0}$[9] is trivial and the OPE is simply given by a single Wilson line associated with $t'_{2,0} \otimes 1 + 1 \otimes \tilde{t}_{2,0}$. Hence, the tree level OPE gives

$$(t'_{2,0} \otimes 1 + 1 \otimes \tilde{t}_{2,0}) \int \partial_z^2 A. \tag{26}$$

On the other hand, the OPE becomes nontrivial at the 1-loop level, as there is an obvious correction coming from the 3-point vertex of the 5d Chern-Simons theory that couples two Wilson lines, as shown in the figure below.

Combining the tree level and the 1-loop level computation, we obtain a single fused Wilson line

$$\left( t'_{2,0} + \tilde{t}_{2,0} + 2\sigma_3 \sum_{m,n \geq 0} \mathrm{d}_{m,n} t'_{0,m} \tilde{t}_{0,n} \tilde{w}^{-m-n-2} \right) \int \partial^2 z A. \tag{27}$$

---

[9]We distinguish two algebra elements in different Wilson lines by prime and tilde.

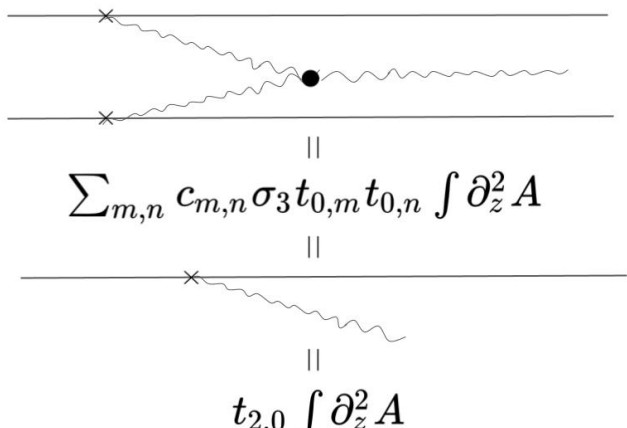

Figure 3: The top figure shows the quantum correction on the Wilson line OPEs from the interaction with the 5d Chern-Simons theory. The formula($\sim \sigma_3 t_{0,m} t_{0,n}$) for the fused Wilson line can be obtained by computing the Feynman diagram. As the representation associated with $\partial_z^2 A$ is $t_{2,0}$, the OPE directly gives the coproduct formula $\Delta_{\mathcal{A},\mathcal{A}} : t_{2,0} \to \dots \sigma_3 t_{0,m} t_{0,n}$.

Since $\int \partial_z^2 A$ couples to $t_{2,0}$ according to (13), the fusion induces an embedding map

$$\Delta_{\mathcal{A},\mathcal{A}} : t_{2,0} \to t'_{2,0} + \tilde{t}_{2,0} + 2\sigma_3 \sum_{m,n \geq 0} \mathrm{d}_{m,n} t'_{0,m} \tilde{t}_{0,n} \tilde{w}^{-m-n-2}. \tag{28}$$

This is exactly (21). As the tree level is trivial, we will only give an explicit derivation of the 1-loop term in §3.4.

We can similarly analyze the surface defect fusion. Given two parallel surface defects, placed on $\mathbb{R}_t$ direction at $t = 0$, $t = -\epsilon$, when we approach them together by taking $\epsilon \to 0$, we obtain a single surface defect. We will directly compute the OPE of two surface defects in the 5d Chern-Simons background using Feynman diagrams.

We will present the nontrivial part of the OPE, which is at 1-loop order, as shown in the figure below. From the 1-loop computation, we obtain a single fused surface defect

$$\dots + \sigma_3 \sum_{n=-\infty}^{\infty} n J_{n-1} J'_{-n-1} \int dw \partial_w (\partial_z^2 A). \tag{29}$$

Since $\int \partial_w A$ couples to $L_{-2}$ according to (17), the fusion induces an embedding map $\Delta_{\mathcal{W}_\infty, \mathcal{W}_\infty}$.

$$L_{-2} \to \dots + \sigma_3 \sum_{n=-\infty}^{\infty} n J_{n-1} J'_{-n-1}. \tag{30}$$

The basic coproduct $\Delta_{\mathcal{W}_\infty, \mathcal{W}_\infty}$ was not explicitly presented in [11], but it was hiding in a composed coproduct $\mathcal{A} \to \mathcal{A} \otimes \mathcal{W}_\infty \otimes \mathcal{W}_\infty$. On the other hand, from [31] we expect the basic coproduct $T \to J \otimes J$, where $T$ is spin-2 current and $J$ is a spin-1 current. (30) is essentially the relevant $\mathcal{O}(\sigma_3)$ order term hiding in the RHS of (2.41) of [11]. We will give a check in §3.6.

## 3.2 Holographic interpretation of the heterotic fusion

We will derive the coproduct $\Delta_{\mathcal{A},\mathcal{W}_\infty}$, based on the gauge invariance of the M2-M5 brane junction configuration. One way to discuss the gauge-invariance of the coupled system is by computing the amplitude of a collection of Feynman diagrams that involve vertices on the defects.

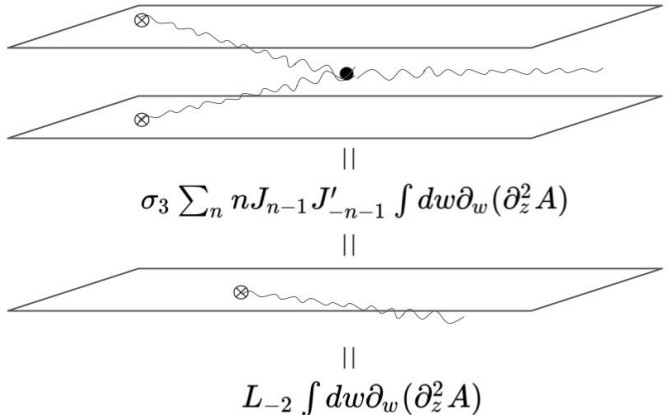

Figure 4: The top figure shows the quantum correction on the surface defect OPEs from the interaction of the two surface defects with the 5d Chern-Simons theory. The formula($\sim \sigma_3 J_{n-1} J'_{-n-1}$) for the fused surface defect can be obtained by computing the Feynman diagram. As the representation associated with $\partial_w A$ is $L_{-2}$, the OPE directly gives the coproduct formula $\Delta_{\mathcal{W}_\infty, \mathcal{W}_\infty} : L_{-2} \to \dots \sigma_3 J_{n-1} J'_{-n-1}$.

To figure out the collection of Feynman diagrams, one needs to consider the LHS(an element of $\mathcal{A}$) of the boundary coproduct relation (25) and determine the 5d gauge mode that would couple to it. The next step is to write down all Feynman diagrams whose amplitude is proportional to the 5d gauge mode.

The LHS of the second line of (25) is $t_{2,0}$ and it couples to $\partial_z^2 A$. The following diagram represents the one associated with the LHS. The amplitude of the Feynman diagram is trivially

$$t_{2,0} \partial_z^2 A. \tag{31}$$

Its BRST variation (14) is

$$t_{2,0} \partial_z^2 (Q_{BRST} A). \tag{32}$$

On the other hand, up to $\mathcal{O}(\sigma_3)$ order, there are three more diagrams, whose amplitudes are proportional to $\partial_z^2 A$. They are The sum of the amplitudes of the Feynman diagram is

$$\left( t_{2,0} + V_{-2} + \sum_{n=1}^\infty n J_{-n-1} t_{0,n-1} \right) \partial_z^2 A. \tag{33}$$

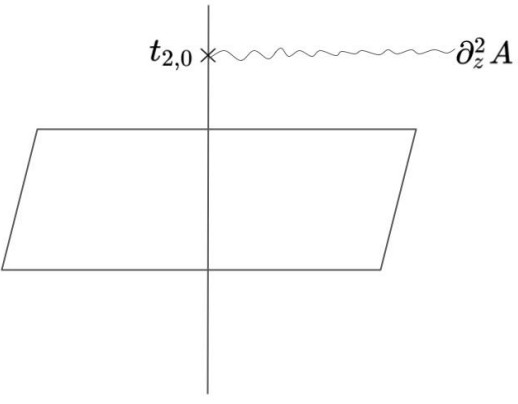

Figure 5: The Feynman diagram associated with the LHS of (25): $t_{2,0}$.

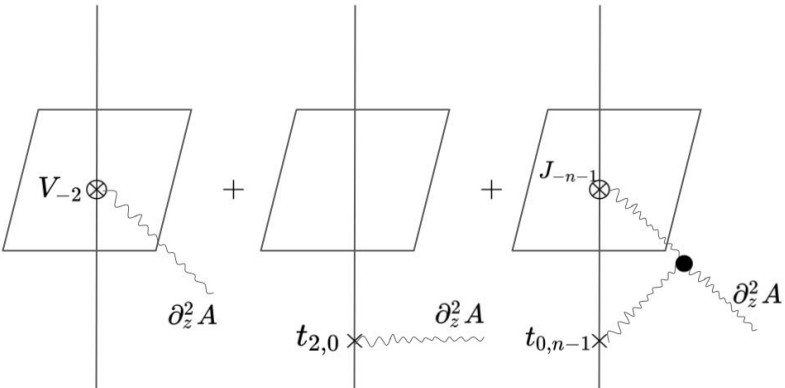

Figure 6: The Feynman diagram associated with the RHS of (25): $t_{2,0} + V_{-2} + \sum_n nJ_{n-1}t_{0,n-1}$.

Its BRST variation (14) is

$$\left( t_{2,0} + V_{-2} + \sum_{n=1}^{\infty} nJ_{-n-1}t_{0,n-1} \right) \partial_z^2 (Q_{BRST}A). \tag{34}$$

For the defect-enriched 5d Chern-Simons theory to be anomaly free, there must be a cancellation between (32) and (34), which leads to the coproduct relation that we have already seen in the second line of (25)[10]:

$$t_{2,0} \rightarrow t_{2,0} + V_{-2} + \sum_{n=1}^{\infty} nJ_{-n-1}t_{0,n-1}. \tag{35}$$

As the tree level $\mathcal{O}(\sigma_3)$ computation is trivial, we will only provide an explicit 1-loop $\mathcal{O}(\sigma_3)$ computation in §3.5.

## 3.3 Ingredients of Feynman diagrams

To prepare for the computation of the Feynman diagrams shown in the previous subsections, we will write down the ingredients of the Feynman diagrams that involve the 5d Chern-Simons theory and two types of defects.

Let us start from the 5d Chern-Simons Lagrangian. From the kinetic term

$$\frac{1}{\sigma_3} dz \wedge dw \wedge A \wedge dA, \tag{36}$$

we can read off the gauge field propagator:

- 5d gauge field propagator $P$ is a solution of

$$dz \wedge dw \wedge dP = \delta_{t=z=w=0}. \tag{37}$$

That is,

$$P_{12} = P(v_1, v_2) = \langle A(v_1)A(v_2) \rangle = \frac{\bar{z}_{12} d\bar{w}_{12} dt_{12} - \bar{w}_{12} d\bar{z}_{12} dt_{12} + t_{12} d\bar{z}_{12} d\bar{w}_{12}}{d_{12}^5}, \tag{38}$$

---

[10]We thank Miroslav Rapčák, who pointed out the previous typos in the following formula.

| Propagator | $V_1 \wedge\wedge\wedge\wedge\sim\sim\sim \overset{P_{12}}{\sim\sim\sim} V_2$ |
| --- | --- |
| 3-point vertex V | $I_{3pt}(V)$ |
| 1-point vertex V | $I_{tA}(V)$ |
| 1-point vertex V | $I_{wA}(V) \otimes\wedge\wedge\sim\sim$ |

Figure 7: A table of ingredients of the Feynman diagrams in the 5d Chern-Simons theory coupled with the line and the surface defects.

where

$$v_i = (t_i, z_i, w_i), \quad d_{ij} = \sqrt{t_{ij}^2 + |z_{ij}|^2 + |w_{ij}|^2},$$
$$t_{ij} = t_i - t_j, \quad z_{ij} = z_i - z_j, \quad w_{ij} = w_i - w_j. \tag{39}$$

From the 3-point coupling

$$\frac{1}{\sigma_3} dz \wedge dw \wedge A \wedge \left( \frac{\partial A}{\partial z} \wedge \frac{\partial A}{\partial w} - \frac{\partial A}{\partial w} \wedge \frac{\partial A}{\partial z} \right), \tag{40}$$

we read off

- Three-point vertex $\mathcal{I}_{3pt}$:

$$\mathcal{I}_{3pt} = \frac{1}{\sigma_3} dz \wedge dw (\partial_z \partial_w). \tag{41}$$

Each of the partial derivatives acts on one of three legs that attaches to the vertex.

From (36), (40), we can see that the loop counting parameter is $\sigma_3$: each of the propagator is proportional to $\sigma_3$ and the internal vertex is proportional to $\sigma_3^{-1}$. Therefore, a given Feynman diagram with $v$ 3-point vertices and $e$ internal propagators is proportional to $\sigma_3^{e-v}$.

Next, consider the line defect coupled to the 5d Chern-Simons theory. Classically, $t_{m,n}$ couples to the mode of the 5d gauge field by

$$\int_{\mathbb{R}} t_{m,n} \partial_z^m \partial_w^n A. \tag{42}$$

From (42), we read off

- One-point vertex $\mathcal{I}_{tA}$:

$$\mathcal{I}_{tA} = \delta_{t,z,w}^{(5)} t_{m,n} \partial_z^m \partial_w^n A. \tag{43}$$

Lastly, consider the surface defect coupled to the 5d Chern-Simons theory. Classically, $W_n^{(m)}$ couples to the mode of the 5d gauge field by

$$\int_{\mathbb{C}_w} W_n^{(m)} \cdot w^{-m-n} \partial_w^{m-1} A dw. \tag{44}$$

From (44), we read off

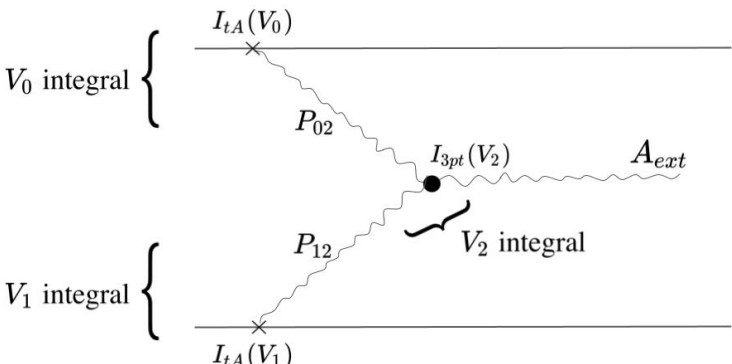

Figure 8: The 1-loop Feynman diagram associated with the $\mathcal{A} \to \mathcal{A} \otimes \mathcal{A}$ coproduct. All the ingredients are explicitly displayed.

- One-point vertex $\mathcal{I}_{wA}$:

$$\mathcal{I}_{wA} = \delta_{t,z}^{(3)} W_n^{(m)} w^{-m-n} \partial_w^{m-1} A dw. \tag{45}$$

As usual in the Feynman diagram computation, we will use (46) in the following subsections, when we evaluate the final integrals.

$$\frac{1}{A^\alpha B^\beta} = \frac{\Gamma(\alpha+\beta)}{\Gamma(\alpha)\Gamma(\beta)} \int_0^1 dx \frac{x^{\alpha-1}(1-x)^{\beta-1}}{(xA+(1-x)B)^{\alpha+\beta}}. \tag{46}$$

Along with it, we used *Mathematica* to compute various integrals; we submitted an ancillary notebook that collects the integral computations.

### 3.4 $\mathcal{A} \to \mathcal{A} \otimes \mathcal{A}$ coproduct

We will derive the meromorphic coproducts of the M2 brane algebra using the perturbative Feynman diagram computation in 5d Chern-Simons theory. The target relation that we want to derive from the 5d Chern-Simons side is

$$t_{2,0} \to \ldots + \sigma_3 \sum_{m,n \geq 0} (\text{const})_{m,n} t_{0,m} t_{0,n} \tilde{z}^{-m-n-2}. \tag{47}$$

We will use the technique developed in [54], where the authors computed the OPE of two Wilson lines using the relevant Feynman diagram in the 4d Chern-Simons theory[11].

Using the ingredients given in §3.3, we can decorate the 1-loop Feynman diagram shown in §3.1 as follows. The amplitude is

$$\sigma_3 t_{0,m} t_{0,n} \int_{V_2} dz_2 dw_2 A_{ext} \int_{V_0} \delta^{(2)}(z_0) \delta^{(2)}(w_0) \partial_{w_0}^m \partial_{z_2} P_{02}$$
$$\times \int_{V_1} \delta^{(2)}(z_1) \delta^{(2)}(w_1 - \tilde{w}) \partial_{w_1}^n \partial_{w_2} P_{12}, \tag{48}$$

where we used (41), (43) for $I_{3pt}(V_2)$, $I_{tA}(V_0)$ and $I_{tA}(V_1)$, respectively.

---

[11]The Yangian coproduct was more explicitly discussed in [55] in the context of the 4d Chern-Simons theory.

There are three floating vertices, so there are three integrals to do. Let us first do $V_0$, $V_1$ integrals and use them in the final $V_2$ integral.

$$\int_{V_0} \delta^{(2)}(z_0)\delta^{(2)}(w_0)\partial_{w_0}^m \partial_{\bar{z}_2} P_{02}. \tag{49}$$

Since $\delta^{(2)}(z_0)\delta^{(2)}(w_0) \sim dz_0 d\bar{z}_0 dw_0 d\bar{w}_0$, we can project most of the terms in $P_{02}$, and get

$$(-1)^m \frac{7}{2}\cdot\frac{9}{2}\cdots\frac{7+2m-2}{2}\int_{V_0}\delta^{(2)}(z_0)\delta^{(2)}(w_0)\bar{w}_2^m \bar{z}_2 \frac{(\bar{z}_2 d\bar{w}_2 - \bar{w}_2 d\bar{z}_2)dt_0}{\sqrt{t_{02}^2 + |w_{02}|^2 + |z_{02}|^2}^{7+2m}}. \tag{50}$$

After shifting $t_0 \to t_0 + t_2$, and evaluating two delta functions, we do the $t_0$ integral. The result is

$$(-1)^m \frac{8\Gamma(3+m)}{15}\frac{\bar{w}_2^m \bar{z}_2(\bar{z}_2 d\bar{w}_2 - \bar{w}_2 d\bar{z}_2)}{(|w_2|^2 + |z_2|^2)^{m+3}}. \tag{51}$$

Now, let us do $V_1$ integral

$$\int_{V_1}\delta^{(2)}(z_1)\delta^{(2)}(w_1 - \tilde{w})\partial_{w_1}^n \partial_{w_2} P_{12}. \tag{52}$$

Taking into account of the $z_1$, $w_1$ delta function, we simplify it into

$$(-1)^n \frac{7}{2}\cdot\frac{9}{2}\cdots\frac{7+2n-2}{2}\int_{-\infty}^{\infty}dt_1 \frac{(\bar{\tilde{w}} - \bar{w}_2)^{n+1}(\bar{z}_2 d\bar{w}_2 + (\bar{\tilde{w}} - \bar{w}_2)d\bar{z}_2)}{\sqrt{t_1^2 + |\tilde{w} - w_2|^2 + |z_2|^2}^{7+2n}}. \tag{53}$$

Doing the $t_1$ integral we get

$$(-1)^m \frac{8\Gamma(3+n)}{15}\frac{(\bar{\tilde{w}} - \bar{w}_2)^{n+1}(\bar{z}_2 d\bar{w}_2 + (\bar{\tilde{w}} - \bar{w}_2)d\bar{z}_2)}{(|\bar{\tilde{w}} - w_2|^2 + |z_2|^2)^{n+3}}. \tag{54}$$

We can then combine (51), (54), and the 3-point interaction vertex $\sigma_3 dz_2 dw_2$, and set up the $V_2$ integral. To be concise, let us omit the constant factors and reintroduce them at the end.

$$\int_{V_2}(\sigma_3 dz_2 dw_2)\frac{\bar{w}_2^m \bar{z}_2(\bar{z}_2 d\bar{w}_2 - \bar{w}_2 d\bar{z}_2)}{(|w_2|^2 + |z_2|^2)^{m+3}}\frac{(\bar{\tilde{w}} - \bar{w}_2)^{n+1}(\bar{z}_2 d\bar{w}_2 + (\bar{\tilde{w}} - \bar{w}_2)d\bar{z}_2)}{(|\bar{\tilde{w}} - w_2|^2 + |z_2|^2)^{n+3}}A_{ext}. \tag{55}$$

We then expand[12] $A_{ext}(z_2)$ in $z_2$ and notice that the only nonvanishing part of the integral comes from one of the modes of $A_{ext}$.

$$A_{ext} = \ldots + z_2^2 \partial_{z_2}^2 A_{ext}. \tag{56}$$

Simplifying the numerator, we get

$$\sigma_3 \int_{V_2}\frac{\bar{w}_2^m(\bar{\tilde{w}} - \bar{w}_2)^{n+1}\bar{z}_2^2 \bar{\tilde{w}}(z_2^2 \partial_{z_2}^2 A)}{(|w_2|^2 + |z_2|^2)^{m+3}(|\tilde{w} - w_2|^2 + |z_2|^2)^{n+3}}|dw_2|^2|dz_2|^2. \tag{57}$$

We can apply Feynman integral (46) here and get

$$\int_0^1 x^{m+2}(1-x)^{n+2}\int_{V_2}\frac{\bar{w}_2^m(\bar{\tilde{w}} - \bar{w}_2)^{n+1}\bar{z}_2^2 \bar{\tilde{w}}(z_2^2 \partial_{z_2}^2 A)|dw_2|^2|dz_2|^2 dx}{((1-x)(|w_2|^2 + |z_2|^2) + x(|\tilde{w} - w_2|^2 + |z_2|^2))^{m+n+6}}. \tag{58}$$

---

[12]See the discussion around equation (3.20) of [54].

We can rewrite the denominator into $(|w_2 - x\tilde{w}|^2 + |z_2|^2 + x(1-x)|\tilde{w}|^2)^{m+n+6}$, and shift $w_2 \to w_2 + x\tilde{w}$. Then, the above becomes

$$\int_0^1 x^{m+2}(1-x)^{n+2} \int_{V_2} \partial_{z_2}^2 A \frac{(\bar{w}_2 + x\bar{\tilde{w}})^m((1-x)\bar{\tilde{w}} - \bar{w}_2)^{n+1}|z_2|^4 \bar{\tilde{w}}}{(|w_2|^2 + |z_2|^2 + x(1-x)|\tilde{w}|^2)^{m+n+6}} |dw_2|^2 |dz_2|^2 dx. \quad (59)$$

When we work in the radial coordinates $(r_z, \theta_z)$, $(r_w, \theta_w)$ on each $\mathbb{C}_z$, $\mathbb{C}_w$ planes, it becomes manifest that all the terms with non-zero powers of $\bar{w}_2$ in the numerator become zero under the $\theta_w$ integral.

Hence, only one term in the expanded numerator survives:

$$\bar{\tilde{w}}^{m+n+2} \int_0^1 x^{2m+2}(1-x)^{2n+3} \int_{V_2} \partial_{z_2}^2 A \frac{|z_2|^4}{(|w_2|^2 + |z_2|^2 + x(1-x)|\tilde{w}|^2)^{m+n+6}} |dw_2|^2 |dz_2|^2 dx. \quad (60)$$

Note that if the external leg were $\partial_{z_2}^n A$ with $n \neq 2$, the amplitude vanishes, and the only non-vanishing condition under $\theta_z$ integral is $n = 2$.

In the radial coordinates, we can evaluate the integral explicitly:

$$\bar{\tilde{w}}^{m+n+2} \int_0^1 x^{2m+2}(1-x)^{2n+3} \int_{\mathbb{R}_t} \partial_{z_2}^2 A \int_0^\infty \int_0^\infty \frac{4\pi^2 r_z^5 r_w}{(r_z^2 + r_w^2 + x(1-x)|\tilde{w}|^2)^{m+n+6}} dr_z dr_w$$

$$= \frac{2\pi^2}{\prod_{i=2}^5 (i+m+n)} \tilde{w}^{-m-n-2} \int_{\mathbb{R}_t} \partial_{z_2}^2 A \int_0^1 x^{m-n}(1-x)^{n-m+1} dx$$

$$= \frac{\pi^2}{\prod_{i=2}^5 (i+m+n)} \tilde{w}^{-m-n-2} \int_{\mathbb{R}_t} \partial_{z_2}^2 A. \quad (61)$$

The integration in the second line converges if and only if $m = n$ or $m = n+1$. Reintroducing the numerical factors that were omitted, we arrive at

$$\sigma_3 \sum_{0 \leq m-n \leq 1} c_{m,n} t_{0,m} t_{0,n} \tilde{w}^{-m-n-2} \int_{\mathbb{R}_t} \partial_{z_2}^2 A, \quad (62)$$

where

$$c_{m,n} = (-1)^{m+n} \left(\frac{8\pi}{15}\right)^2 (m+n+1)!. \quad (63)$$

We have obtained a single composite Wilson line associated with the tensor product representation $t_{0,m} \otimes t_{0,n} \in \mathcal{A} \otimes \mathcal{A}$. Due to the coupling (13), the tensor product representation can be equally understood as $t_{2,0} \in \mathcal{A}$. Therefore, we have derived the 1-loop part of the seed coproduct relation of $\Delta_{\mathcal{A},\mathcal{A}} : \mathcal{A} \to \mathcal{A} \otimes \mathcal{A}$.

$$t_{2,0} \to \ldots + \sigma_3 \sum_{0 \leq m-n \leq 1} c_{m,n} t'_{0,m} \tilde{t}_{0,n} w^{-m-n-2}. \quad (64)$$

We should emphasize that although we have spent most of the space to compute the integral, it is only for checking and showing that the integral converges to a finite quantity for a particular component of the expansion of $A_{ext}$ (56). More emphasis should be placed on the selection rule that determines which structure constants to vanish or not. In the present case, the selection rule restricts the RHS of the coproduct to have only $t'_{0,m}\tilde{t}_{0,n}$.

One can still compare the structure constant $c_{m,n}$ in (64) and its Koszul dual structure constant $d_{m,n}$ in (21). In general, we do not expect a precise equality between two; there can

be overall numerical factor. For instance, let us recall [54], where the author compared the OPE in $C^*(\mathbb{C}_{\epsilon_2}[z_1, z_2] \otimes \mathfrak{gl}_1)$

$$\{\partial_{z_1}^p \partial_{z_2}^q X, \partial_{z_1}^k \partial_{z_2}^l X\} = \sum \epsilon_1 \epsilon_2^{(r+s+m+n-p-q-k-l)/2-1} A_{r,s,m,n}^{p,q,k,l} (\partial_{z_1}^r \partial_{z_2}^s X)(\partial_{z_1}^m \partial_{z_2}^n X), \qquad (65)$$

where $\partial_{z_1}^p \partial_{z_2}^q X \in C^*(\mathbb{C}_{\epsilon_2}[z_1, z_2] \otimes \mathfrak{gl}_1)$, and the Koszul dual OPE in $U(\mathbb{C}_{\epsilon_2}[z_1, z_2]\mathbb{C} \otimes \mathfrak{gl}_1)$

$$[z_1^r z_2^s, z_1^m z_2^n] = \sum \epsilon_1 \epsilon_2^{(r+s+m+n-p-q-k-l)/2-1} A_{r,s,m,n}^{p,q,k,l} \frac{m!n!r!s!}{p!q!k!l!} (z_1^p z_2^q)(z_1^k z_2^l), \qquad (66)$$

where $z_1^m z_2^n \in U(\mathbb{C}_{\epsilon_2}[z_1, z_2]\mathbb{C} \otimes \mathfrak{gl}_1)$. The analogue of $d_{m,n}$ in (21) is the structure constant $A_{r,s,m,n}^{p,q,k,l}$ that appears in (65) and the analogue of $c_{m,n}$ in (64) is the structure constant $A_{r,s,m,n}^{p,q,k,l}$ multiplied by the numerical factor that follows. In this case, the structure constants of Koszul dual pair algebra are related the numerical constant.

### 3.5 $\mathcal{A} \to \mathcal{A} \otimes \mathcal{W}_\infty$ coproduct

We will derive the coproducts of M2 brane algebra and M5 brane algebra using the perturbative Feynman diagram computation in 5d Chern-Simons theory. The target relation that we want to derive from the 5d Chern-Simons side is

$$t_{2,0} \to \dots + \sigma_3 \sum_{n=1}^\infty n W_{-n-1}^{(1)} W_{n-1}^{(1)} + \sigma_3 \sum_{n=1}^\infty n W_{-n-1}^{(1)} t_{0,n-1}. \qquad (67)$$

This situation is similar to the intersecting M2-M5 brane configuration studied in [7, 9]. To derive the coproduct relation holographically, we will follow [9], where we computed the Feynman diagrams involving a line and a surface defect.

Let us first write the RHS of (67) in the manifest form of $\mathcal{A} \otimes \mathcal{W}_\infty$, by recalling the embedding $\rho(J_{n-1}) = t_{0,n}$.

$$\sigma_3 \sum_{n=1}^\infty n J_{-n-1} J_{n-1} + \sigma_3 \sum_{n=1}^\infty n J_{-n-1} t_{0,n-1} = \sigma_3 \sum_{n=1}^\infty n J_{-n-1} t_{0,n-1}. \qquad (68)$$

Using the ingredients given in §3.3, we can decorate the 1-loop Feynman diagram shown in §3.2 as follows.

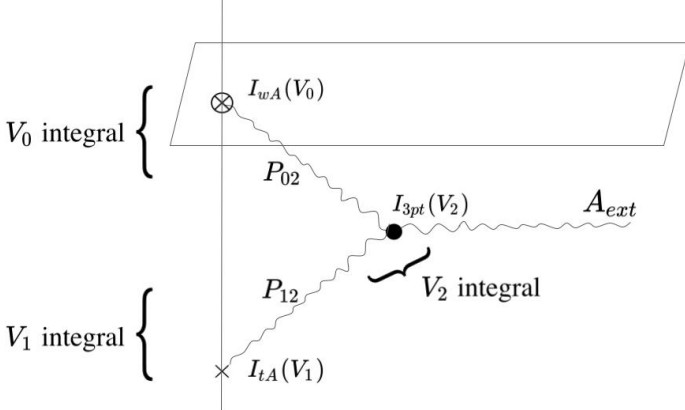

Figure 9: The 1-loop Feynman diagram associated with the $\mathcal{A} \to \mathcal{A} \otimes \mathcal{W}_\infty$ coproduct. All the ingredients are explicitly displayed.

The amplitude is

$$
\sigma_3 J_{-n-1} t_{0,n-1} \int_{V_2} A_{ext} dz_2 dw_2 \int_{V_0} \delta(t_0)\delta^{(2)}(z_0)\partial_{\bar{z}_2}(w_0^n P_{02}) dw_0
$$
$$
\times \int_{V_1} \delta(t_1-\epsilon)\delta^{(2)}(z_1)\delta^{(2)}(w_1)\partial_{w_2}\partial_{w_1}^{n-1} P_{12}, \tag{69}
$$

where we used (41), (44), (43) for $I_{3pt}(V_2)$, $I_{wA}(V_0)$ and $I_{tA}(V_1)$, respectively.

Let us omit all constant terms and reintroduce them at the end. There are three floating vertices, so there are three integrals to do. Let us first do $V_0$, $V_1$ integrals and use them in the final $V_2$ integral.

$$
\int_{V_0} \delta(t_0)\delta^{(2)}(z_0)w_0^n \partial_{\bar{z}_2} P_{02}. \tag{70}
$$

Since $\delta(t_0)\delta^{(2)}(w_0) \sim dt dw_0 d\bar{w}_0$, we can project most of the terms in $P_{02}$. After performing $t_0$, $z_0$ integral, we get

$$
-\int_{\mathbb{C}_{w_0}} \frac{w_0^n \bar{z}_2(\bar{z}_2 dt_2 + t_2 d\bar{z}_2)}{\sqrt{t_2^2 + |z_2|^2 + |w_{02}|^2}^7} |dw_0|^2. \tag{71}
$$

After shifting $w_0 \rightarrow w_0 + w_2$, we get

$$
-\int_{\mathbb{C}_{w_0}} \frac{\bar{z}_2(w_0 + w_2)^n(t_2 d\bar{z}_2 + \bar{z}_2 dt_2)}{\sqrt{t_2^2 + |z_2|^2 + |w_0|^2}^7} |dw_0|^2. \tag{72}
$$

Working in the radial coordinate of $\mathbb{C}_{w_0}$ plane, only one term in the expanded $(w_0 + w_2)^n$ survives. Performing the $w_0$ integral in the radial coordinate, we get

$$
-\frac{2\pi}{3} \frac{w_2^n \bar{z}_2}{\sqrt{t_2^2 + |z_2|^2}^5}(t_2 d\bar{z}_2 + \bar{z}_2 dt_2). \tag{73}
$$

Now, let us do $V_1$ integral.

$$
\int_{V_1} \delta(t_2+\epsilon)\delta^{(2)}(z_1)\delta^{(2)}(w_1)\partial_{w_1}^{n-1}\partial_{w_2} P_{12}. \tag{74}
$$

As there are 3 delta functions, we can easily get

$$
\left((-1)^n \frac{7}{2} \cdot \frac{9}{2} \cdots \frac{7+2n-2}{2}\right) \frac{-\bar{w}_2^n(-\bar{z}_2 d\bar{w}_2 dt_2 + \bar{w}_2 d\bar{z}_2 dt_2 - (\epsilon+t_2)d\bar{z}_2 d\bar{w}_2)}{\sqrt{(\epsilon+t_2)^2 + |z_2|^2 + |w_2|^2}^{5+2n}}. \tag{75}
$$

The numerical factor in front can be written in terms of $\Gamma$ function and will be incorporated later in the final formula.

We can then combine (73), (75), and the 3-point interaction vertex $\sigma_3 dz_2 dw_2$, and set up the $V_2$ integral.

$$
\sigma_3 \int_{V_2} dz_2 dw_2 \frac{|w_2|^{2n}\bar{z}_2(\bar{z}_2 dt_2 + t_2 d\bar{z}_2)(-\bar{z}_2 d\bar{w}_2 dt_2 + \bar{w}_2 d\bar{z}_2 dt_2 - (\epsilon+t_2)d\bar{z}_2 d\bar{w}_2)}{\sqrt{t_2^2 + |z_2|^2}^5 \sqrt{|\epsilon+t_2|^2 + |w_2|^2 + |z_2|^2}^{5+2n}} A_{ext}. \tag{76}
$$

We then expand[13] $A_{ext}(z_2)$ in $z_2$ and notice that the only nonvanishing part of the integral comes from one of the modes of $A_{ext}$.

$$
A_{ext} = \ldots + z_2^2 \partial_{z_2}^2 A_{ext}. \tag{77}
$$

---

[13]See the discussion around equation (3.20) of [54].

Substituting it in and simplifying the numerator, we get

$$\sigma_3 \int_{V_2} \frac{|w_2|^{2n}|z_2|^4(\epsilon + 2t_2)}{\sqrt{t_2^2 + |z_2|^2}^5 \sqrt{(\epsilon + t_2)^2 + |w_2|^2 + |z_2|^2}^{5+2n}} dt_2 |dw_2|^2 |dz_2|^2. \tag{78}$$

Note that if the external leg were not $z_2^2 \partial_{z_2}^2 A$, but $z_2^n \partial_{z_2}^n A$ with $n \neq 2$, the $z_2$-integral would vanish.

We can now apply Feynman integral (46) on (78). Omitting $\Gamma$ functions for now, and setting $\epsilon = 1$, we get

$$\sigma_3 \int_0^1 dx \int_{V_2} \frac{\sqrt{x^{2n+3}(1-x)^3}|w_2|^{2n}|z_2|^4(2t_2+1)}{((1-x)(t_2^2 + |z_2|^2) + x(|w_2|^2 + |z_2|^2 + (1+t_2)^2))^{n+5}} |dw_2|^2 |dz_2|^2 dt_2. \tag{79}$$

We can rewrite the denominator into $(|z_2|^2 + x|w_2|^2 + (t_2 + x)^2 + x(1-x))^{m+n+6}$, and work in radial coordinates $(r_z, \theta_z)$, $(r_w, \theta_w)$ for both $\mathbb{C}_z$, $\mathbb{C}_w$ planes. Then, the above becomes

$$4\pi^2\sigma_3 \int_0^1 dx \sqrt{x^{2n+3}(1-x)^3} \int \frac{r_w^{2n+1} r_z^5 (2t_2+1)}{(r_z^2 + xr_w^2 + (t_2+x)^2 + x(1-x))^{n+5}}. \tag{80}$$

Then, shift $t_2 \to t_2 - x$, and rescale $r_w \to r_w/\sqrt{x}$. Using the fact that the integral domain for $t_2$ is $(-\infty, \infty)$, a term with an odd power of $t_2$ vanishes.

$$4\pi^2\sigma_3 \int_0^1 dx \sqrt{x^3(1-x)^3}(1-2x) \int_0^\infty dr_z \int_0^\infty dr_w \int_{-\infty}^\infty dt_2 \frac{r_w^{2n+1} r_z^5}{(r_w^2 + r_z^2 + t_2^2 + x(1-x))^{n+5}}. \tag{81}$$

The final integral is straightforward to evaluate and it gives

$$\sigma_3 \frac{\pi^4}{256} \frac{\Gamma(1+n)}{\Gamma(5+n)}. \tag{82}$$

Re-introducing all omitted numerical factors, we arrive at

$$\sum_n c_n \sigma_3 J_{-n-1} t_{0,n-1} \partial_{z_2}^2 A, \tag{83}$$

where

$$c_n = \frac{\pi^4}{144} \frac{n!}{2n+5}. \tag{84}$$

As it has the external leg $\partial_{z_2}^2 A$, this Feynman diagram mixes with Figure 5 and the first two of Figure 6. The BRST variation (14) of these Feynman diagrams should sum to zero for anomaly-free coupled systems. Hence, we recover the desired coproduct relation.

$$t_{2,0} \to t_{2,0} + \sigma_3 \sum_{n=1}^\infty (const) J_{-n-1} t_{0,n-1} + \dots. \tag{85}$$

We should emphasize that although we have spent most of the space to compute the integral, it is only for checking and showing that the integral converges to a finite quantity for a particular component of the expansion of $A_{ext}$ (77). More emphasis should be placed on the selection rule that determines which structure constants to vanish or not. In the present case, the selection rule restricts the RHS of the coproduct to have only $J_{-n-1} t_{0,n-1}$. Similar remark that we made at the end of §3.4 applies for the numerical part of the structure constant.

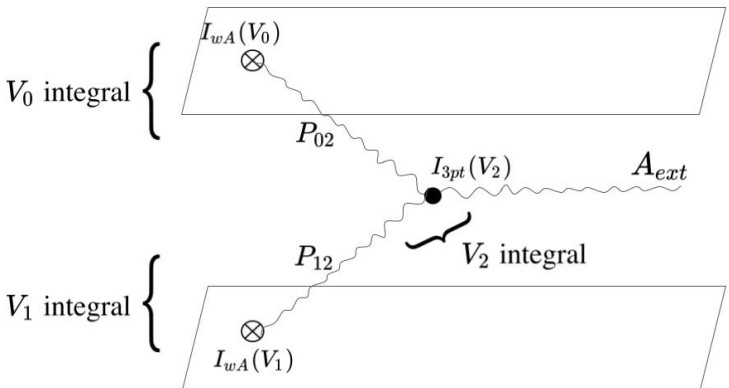

Figure 10: The 1-loop Feynman diagram associated with the $\mathcal{W}_\infty \to \mathcal{W}_\infty \otimes \mathcal{W}_\infty$ coproduct. All the ingredients are explicitly displayed.

### 3.6  $\mathcal{W}_\infty \to \mathcal{W}_\infty \otimes \mathcal{W}_\infty$ coproduct

We will derive the coproducts of M5 brane algebra using the perturbative Feynman diagram computation in 5d Chern-Simons theory. The target relation that we want to derive from the 5d Chern-Simons side is

$$L_{-2} \to \ldots + \sigma_3 \sum_{n=-\infty}^{\infty} n J_{n-1} J'_{-n-1}. \tag{86}$$

We will use the technique developed in [54], where the authors computed the OPE of two Wilson lines by computing the relevant Feynman diagram in the 4d Chern-Simons theory.

Using the ingredients given in §3.3, we can decorate the 1-loop Feynman diagram shown in §3.1 as follows. The amplitude is

$$\sigma_3 J_{n-1} J'_{-n-1} \int_{V_2} dz_2 dw_2 (w_2 \partial_{w_2} A_{\text{ext}}) \int_{V_0} \delta(t_0) \delta^{(2)}(z_0) dw_0 \partial_{z_2}(w_0^n P_{02})$$
$$\times \int_{V_1} \delta(t_1 - (-\epsilon)) \delta^{(2)}(z_1) dw_1 \partial_{w_2}(w_1^{-n} P_{12}), \tag{87}$$

where we used (41), (44) for $I_{3pt}(V_2)$, $I_{wA}(V_0)$ and $I_{wA}(V_1)$, respectively. Here, we started by inserting the explicit term of the expansion $A_{ext} = \ldots + w_2 \partial_{w_2} A_{ext} + \ldots$ in (87), as the LHS of (86) tells us that the external leg should be proportional to $\partial_{w_2} A_{ext}$. Hence, the computation in this subsection should be thought of as a check, not a derivation.

There are three floating vertices, so there are three integrals to do. Let us first do $V_0$, $V_1$ integrals and use them in the final $V_2$ integral.

$$\int_{V_0} \delta(t_0) \delta^{(2)}(z_0) w_0^n \partial_{z_2} P_{02}. \tag{88}$$

Since $\delta(t_0) \delta^{(2)}(z_0) \sim dt_0 dz_0 \bar{z}_0$, we can project most of the terms in $P_{02}$, and get

$$-\frac{5}{2} \int_{\mathbb{C}_{w_0}} \frac{w_0^n \bar{z}_2 (\bar{z}_2 dt_2 + t_2 d\bar{z}_2)}{\sqrt{t_2^2 + |z_2|^2 + |w_{02}|^2}^7} |dw_0|^2. \tag{89}$$

After shifting $w_0 \to w_0 + w_2$, expanding $(w_0 + w_2)^n$ in the numerator, and working in the radial

coordinates $(r_w, \theta_w)$ of $\mathbb{C}_{w_0}$, we can project everything but one term:

$$-\frac{5}{2}\int_0^\infty \frac{w_2^n \bar{z}_2(t_2 d\bar{z}_2 + \bar{z}_2 dt_2)}{\sqrt{t_2^2 + |z_2|^2 + r_w^2}^7}(2\pi r_w dr_w). \tag{90}$$

Doing $r_w$ integral, we have

$$-\pi\frac{w_2^n \bar{z}_2(t_2 d\bar{z}_2 + \bar{z}_2 dt_2)}{\sqrt{t_2^2 + |z_2|^2}^5}. \tag{91}$$

Now, let us do $V_1$ integral.

$$\int_{V_1} \delta(t_1 + \epsilon)\delta^{(2)}(z_1)(w_1)^{-n}\partial_{w_2}P_{12}. \tag{92}$$

Taking into account of the $t_1$, $w_1$ delta function, we simplify it into

$$\frac{5}{2}\int \frac{w_1^{-n}(\bar{z}_2 dt_2 + (t_2 + \epsilon)d\bar{z}_2)\bar{w}_{12}}{\sqrt{(t_2 + \epsilon)^2 + |z_2|^2 + |w_{12}|^2}^7}|dw_1|^2. \tag{93}$$

Shifting $w_1 \to w_1 + w_2$, and expanding $(w_1 + w_2)^{-n}$ in $w_1/w_2$ and in $w_2/w_1$ respectively in the region of convergence, we have

$$\frac{5}{2}\int_{0\leq|w_1|\leq|w_2|} \frac{\bar{w}_1 w_2^{-n}\left(1 - n\frac{w_1}{w_2} + \dots\right)(\bar{z}_2 dt_2 + (t_2 + \epsilon)d\bar{z}_2)}{\sqrt{(t_2 + \epsilon)^2 + |z_2|^2 + |w_1|^2}^7}|dw_1|^2$$
$$+ \frac{5}{2}\int_{|w_2|\leq|w_1|<\infty} \frac{\bar{w}_1 w_1^{-n}\left(1 - n\frac{w_2}{w_1} + \dots\right)(\bar{z}_2 dt_2 + (t_2 + \epsilon)d\bar{z}_2)}{\sqrt{(t_2 + \epsilon)^2 + |z_2|^2 + |w_1|^2}^7}|dw_1|^2. \tag{94}$$

In the radial coordinates $(r_w, \theta_w)$ of $\mathbb{C}_{w_1}$, it is clear that only one term in the expansion in the first parenthesis survives, and reduces to

$$\frac{5}{2}\int_0^{|w_2|} \frac{-nw_2^{-n-1}r_w^2(\bar{z}_2 dt_2 + (t_2 + \epsilon)d\bar{z}_2)}{\sqrt{(t_2 + \epsilon)^2 + |z_2|^2 + r_w^2}^7}(2\pi r_w)dr_w. \tag{95}$$

Doing the $r_w$ integral, we get

$$-\frac{\pi nw_2^{-n-1}(\bar{z}_2 dt_2 + (t_2 + \epsilon)d\bar{z}_2)}{3}\left(\frac{1}{\sqrt{(t_2 + \epsilon)^2 + |z_2|^2}^3} + \frac{2((t_2 + \epsilon)^2 + |z_2|^2) + 5|w_2|^2}{\sqrt{(t_2 + \epsilon)^2 + |z_2|^2 + |w_2|^2}^5}\right). \tag{96}$$

We can then combine (91), (96), the 3-point interaction vertex $\sigma_3 dz_2 dw_2$, and the external leg $A$. This sets up the $V_2$ integral. To be concise, let us omit the constant factors and reintroduce them at the end.

$$\sigma_3 \int dw_2 dt_2 |dz_2|^2 \frac{\bar{z}_2^2(2t_2 + \epsilon)\partial_{w_2}A}{\sqrt{t_2^2 + |z_2|^2}^5}\left(\frac{1}{\sqrt{(t_2 + \epsilon)^2 + |z_2|^2}^3} + \frac{2((t_2 + \epsilon)^2 + |z_2|^2) + 5|w_2|^2}{\sqrt{(t_2 + \epsilon)^2 + |z_2|^2 + |w_2|^2}^5}\right). \tag{97}$$

We may further expand[14] $\partial_{w_2}A(z_2)$ and notice the only nonvanishing piece comes from

$$\partial_{w_2}A = \dots + z_2^2 \partial_{z_2}^2(\partial_{w_2}A). \tag{98}$$

---

[14]See the discussion around equation (3.20) of [54].

Substituting it in and simplifying the integral, we have

$$\sigma_3 \int dw_2(\partial_{w_2}(\partial_{z_2}^2 A)) \int dt_2 |dz_2|^2 \frac{|z_2|^4(2t_2+\epsilon)}{\sqrt{t_2^2+|z_2|^2}^5} \left( \frac{1}{\sqrt{(t_2+\epsilon)^2+|z_2|^2}^3} \right.$$
$$\left. + \frac{2((t_2+\epsilon)^2+|z_2|^2)+5|w_2|^2}{\sqrt{(t_2+\epsilon)^2+|z_2|^2+|w_2|^2}^5} \right). \tag{99}$$

Let us apply Feynman integral (46) to each of two terms, omitting $\Gamma$ functions for now, and setting $\epsilon = 1$. For the first term, we get

$$\sigma_3 \int dw_2(\partial_{w_2}(\partial_{z_2}^2 A)) \int_0^1 dx \sqrt{(1-x)^3 x} \int \frac{|z_2|^4(2t_2+1)dt_2|dz_2|^2}{((1-x)(t_2^2+|z_2|^2)+x((t_2+1)^2+|z_2|^2))^4}. \tag{100}$$

We can rewrite the denominator into $((t_2+x)^2+|z_2|^2+x(1-x))^4$, and shift $t_2 \to t_2-x$. Since the $t_2$-integral domain is $(-\infty, \infty)$, the $t_2$-linear term vanishes. Then, the above becomes

$$\sigma_3 \int dw_2(\partial_{w_2}(\partial_{z_2}^2 A)) \int_0^1 dx \sqrt{(1-x)^3 x} \int dt_2 |dz_2|^2 \frac{|z_2|^4(1-2x)}{(t_2^2+|z_2|^2+x(1-x))^4}. \tag{101}$$

Working in radial coordinates $(r_z, \theta_z)$ on $\mathbb{C}_{z_2}$ plane, we can perform the integral straightforwardly as

$$\frac{\pi}{36}\sigma_3 \int dw_2 \partial_{w_2}(\partial_{z_2}^2 A). \tag{102}$$

Similarly, for the second term of (99), we apply Feynman integral (46).

$$\sigma_3 \int dw_2(\partial_{w_2}(\partial_{z_2}^2 A)) \int_0^1 dx \sqrt{(1-x)^5 x^3}$$
$$\times \int \frac{|z_2|^4(2t_2+1)(2((t_2+1)^2+|z_2|^2)+5|w_2|^2)dt_2|dz_2|^2}{((1-x)(t_2^2+|z_2|^2)+x((t_2+1)^2+|z_2|^2+|w_2|^2))^5}. \tag{103}$$

We can rewrite the denominator into $((t_2+x)^2+|z_2|^2+x|w_2|^2+x(1-x))^4$, shift $t_2 \to t_2-x$, and re-scale $w_2 \to w_2/\sqrt{x}$. Since the $t_2$-integral domain is $(-\infty, \infty)$, the $t_2$-linear term vanishes. Then, the above becomes

$$\sigma_3 \int dw_2(\partial_{w_2}(\partial_{z_2}^2 A)) \int_0^1 dx \sqrt{(1-x)^5} x^2$$
$$\times \int dt_2 |dz_2|^2 \frac{|z_2|^4(1-2x)(2(t_2^2+(1-x)^2+|z_2|^2)+5|w_2|^2/x)}{(t_2^2+|z_2|^2+|w_2|^2+x(1-x))^4}. \tag{104}$$

Working in the radial coordinates $(r_z, \theta_z)$ of $\mathbb{C}_{z_2}$ plane, we can check all the terms in the integrand nicely converge under the $r_z$, $t_2$, $x$ integrals and (104) evaluate to

$$\sigma_3 \int dw_2 \partial_{w_2}(\partial_{z_2}^2 A) \left( \frac{\pi}{256} + \frac{\pi}{48}\left( \frac{3092}{3465} + 4|w_2|^2 \right) + \frac{\pi^2}{12288} \right). \tag{105}$$

As we are working in the holomorphic supergravity background in the $\mathbb{C}_w$ direction, the term $|w_2|^2 = w_2 \bar{w}_2$ with an extra anti-holomorphic dependence on $\bar{w}_2$ must be Q-exact, and we may safely drop it.

Combining (102) and (105), and re-introducing all the omitted constant factors, we arrive at[15]

$$\sigma_3 \, n J_{n-1} J'_{-n-1}(\text{const}) \int dw_2 \partial_{w_2}(\partial_{z_2}^2 A), \tag{106}$$

where

$$(\text{const}) = \frac{\pi^2}{3} \frac{\Gamma(4)}{\Gamma\left(\frac{5}{2}\right)\Gamma\left(\frac{3}{2}\right)} \frac{\Gamma(5)}{\Gamma\left(\frac{5}{2}\right)\Gamma\left(\frac{5}{2}\right)} \left( \frac{73\pi}{2304} + \frac{\pi}{48}\left(\frac{3092}{3465}\right) + \frac{\pi^2}{12288} \right). \tag{107}$$

We have obtained a single composite surface operator associated to the tensor product representation $J_{n-1} \otimes J'_{-n-1} \in \mathcal{W}_\infty \otimes \mathcal{W}_\infty$. Let us look at the external leg $\partial_{w_2}\partial_{z_2}^2 A$, and recall the coupling (17). Since it only tells us about the coupling between the $w_2$ modes of the 5d gauge field and $\mathcal{W}_\infty$ modes, we do not understand what $\partial_{z_2}^2$ means in terms of Koszul duality. Focusing on $\partial_{w_2}A$, as it couples to $L_{-2}$, the tensor product representation $J_{n-1} \otimes J'_{-n-1}$ can be equally understood as $L_{-2} \in \mathcal{W}_\infty$; it induces the coproduct.

Therefore, we have derived the 1-loop part of the basic coproduct relation of $\Delta_{\mathcal{W}_\infty, \mathcal{W}_\infty} : \mathcal{W}_\infty \to \mathcal{W}_\infty \otimes \mathcal{W}_\infty$

$$L_{-2} \to \ldots + \sigma_3 \sum_{n \geq 1} (\text{const}) n J_{n-1} J'_{-n-1}. \tag{108}$$

This is the expected coproduct formula for $\mathcal{W}_{2,0,0} \to \mathcal{W}_{1,0,0} \otimes \mathcal{W}_{1,0,0}$. $\mathcal{W}_{2,0,0}$ is a direct sum of a Virasoro algebra, which provides the mode $L_{-2}$, and an affine Kac-Moody algebra $\hat{\mathfrak{u}}(1)$. $\mathcal{W}_{1,0,0}$ is an affine Kac-Moody algebra $\hat{\mathfrak{u}}(1)$, according to [31].

## 3.7 A comment on the fusion of transverse surface defects

In this section we consider a pair of transverse holomorphic surface defects. Since there is a $\mathrm{SL}_2(\mathbb{C})$ symmetry, we can assume that this pair of surface defects are supported on $\mathbb{C}_z$ and $\mathbb{C}_w$ respectively.

We conjecture that a fusion of two transverse surface defects will give a line operator as a quantum correction in 1-loop order, along with the transverse surface operators. Since we do not have a candidate field theory result for the transverse surface defect fusion, we will not specify a particular mode of $\mathcal{W}_\infty$ algebra that would appear in the coproduct in this subsection. We already have all the ingredients of this calculation. We will frequently draw them from the previous subsections.

We would like to compute the 1-loop correction to the OPE between two transverse surface defects. Diagrammatically, it is given by the following figure.

The amplitude is schematically

$$\sigma_3(\ldots) \int_{V_2} dz_2 dw_2 A_{ext} \int_{V_0} \delta(t_0)\delta^{(2)}(z_0) dw_0 \partial_{z_2}(\ldots P_{02})$$

$$\times \int_{V_1} \delta(t_1)\delta^{(2)}(z_1) dw_1 \partial_{w_2}(\ldots P_{12}), \tag{109}$$

where $\ldots$'s depend on the detail of the modes of the $\mathcal{W}_\infty$ on each of the vertices $I_{wa}(V_0)$ and $I_{zA}(V_1)$. Since the omitted parts do not affect the structural result that we claim, we will not specify those throughout this subsection.

---

[15]It is unclear how to intrerpret $\partial_{z_2}^2$ acting on $A$, as the coupling does not give any information on the $z$ coordinate, but just modes in $\mathbb{C}_w$ plane.

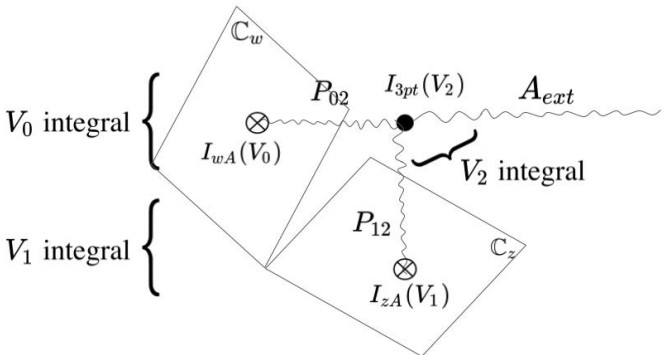

Figure 11: The 1-loop Feynman diagram for the OPE between two transverse surface defects on $\mathbb{C}_z$ and $\mathbb{C}_w$ planes. All the ingredients are explicitly displayed.

As we have learned how to do the integral along the surface defect in the previous subsection, for each $V_0$, $V_1$ integral, we will draw the result from there:

$$
\begin{aligned}
\int_{V_0} \delta(t_0)\delta^{(2)}(z_0)(\ldots)\partial_{z_2}P_{02} &= (\text{const})\frac{(\ldots)\bar{z}_2(t_2d\bar{z}_2+\bar{z}_2dt_2)}{\sqrt{t_2^2+|z_2|^2}^5}, \\
\int_{V_1} \delta(t_1)\delta^{(2)}(z_1)(\ldots)\partial_{w_2}P_{12} &= (\text{const})\frac{(\ldots)\bar{w}_2(t_2d\bar{w}_2+\bar{w}_2dt_2)}{\sqrt{t_2^2+|w_2|^2}^5}.
\end{aligned}
\tag{110}
$$

We can then combine (110) and the 3-point interaction vertex $\sigma_3 dz_2 dw_2$, and the external leg $A_{ext}$. This sets up the $V_2$ integral.

$$
\sigma_3 \int_{V_2} dz_2 dw_2 \frac{(\ldots)\bar{z}_2(t_2d\bar{z}_2+\bar{z}_2dt_2)}{\sqrt{t_2^2+|z_2|^2}^5}\frac{(\ldots)\bar{w}_2(t_2d\bar{w}_2+\bar{w}_2dt_2)}{\sqrt{t_2^2+|w_2|^2}^5}A_{ext}.
\tag{111}
$$

Expanding the numerator, we can observe three objects with a $\sigma_3$ factor omitted.

$$
\begin{aligned}
\int (\ldots)A_{ext}dw_2 \int \frac{(\ldots)dz_2d\bar{z}_2dt_2}{\sqrt{(t_2^2+|z_2|^2)(t_2^2+|w_2|^2)}^5} + \int (\ldots)A_{ext}dz_2 \int \frac{(\ldots)dw_2d\bar{w}_2dt_2}{\sqrt{(t_2^2+|z_2|^2)(t_2^2+|w_2|^2)}^5} \\
+ \int (\ldots)A_{ext} \int \frac{(\ldots)dz_2d\bar{z}_2dw_2d\bar{w}_2}{\sqrt{(t_2^2+|z_2|^2)(t_2^2+|w_2|^2)}^5}.
\end{aligned}
\tag{112}
$$

Depending on $(\ldots)$ in the numerators, combined with a proper term from the expansion of $A_{ext}$ in $z$ or $w$, each integral may or may not produce non-zero answers. As our primary purpose is to see the structure, let us now assume that each integral gives a nonzero answer.

The second integrals in each term evaluate to finite constants, which we again denote by the uniform format $(\ldots)$.

$$
\sigma_3(\text{const})\int_{\mathbb{C}_w} (\ldots)A_{ext}dw_2 + \sigma_3(\text{const})\int_{\mathbb{C}_z} (\ldots)A_{ext}dz_2 + \sigma_3(\text{const})\int_{\mathbb{R}_t} (\ldots)A_{ext}.
\tag{113}
$$

Therefore, in the most general case, we would obtain either a surface operator on $\mathbb{C}_w$, surface operator on $\mathbb{C}_z$, or a line operator on $\mathbb{R}_t$ as a result of the fusion of two transverse surface defects, especially from the 1-loop quantum correction.

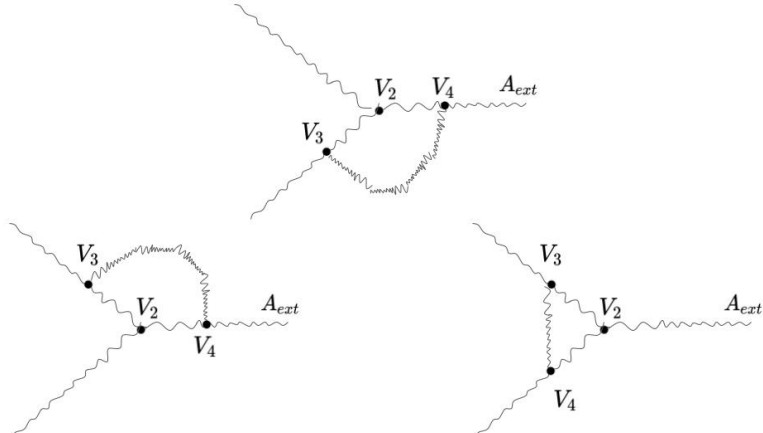

Figure 12: Three possible higher loop corrections. In addition to the already existed vertices $V_0$, $V_1$, $V_2$ that were used in the computation in the previous subsections, it contains two extra vertices $V_3$, $V_4$, and three extra propagators $P_{32}$, $P_{34}$, $P_{24}$. We distinguish the internal propagators and the external leg, by showing $A_{ext}$ explicitly on the external leg.

## 3.8    1-loop exactness of the Feynman diagrams

All basic coproducts $\Delta_{\mathcal{A},\mathcal{A}}$, $\Delta_{\mathcal{A},\mathcal{W}_\infty}$, $\Delta_{\mathcal{W}_\infty,\mathcal{W}_\infty}$ that we have tried to reproduce so far truncate at $\mathcal{O}(\sigma_3)$ order [11]. However, in principle, all the diagrams that we have discussed may have higher loop corrections on the internal 3-point vertex and one of the propagators. To match with the algebraic result of [11], we need to argue that such higher corrections vanish. Note that [55] showed the 1-loop exactness of Yangian coproduct using the 4d Chern-Simons Feynman diagrams.

Let us start with the potential higher loop corrections to the internal 3-point vertex. Two new internal vertices $V_3$, $V_4$ and three new internal propagators $P_{32}$, $P_{34}$, $P_{24}$ will introduce an extra factor of $\sigma_3 = \sigma_3^{3-2}$. Hence, with these further corrections, the diagrams, presented in §3, are proportional to $\mathcal{O}(\sigma_3^2)$.

We will argue the vanishing of the higher loop corrections without introducing complicated integrals again since we have learned the rule of the game from the 1-loop computations in the previous subsections. For simplicity, we will focus on the left corner diagram, but the other diagrams are equivalent, as it will turn out soon.

By (41), each of the new 3-point vertices $V_3$, $V_4$ will introduce $\mathcal{I}_{3pt}(V_i)$: a factor of $\sigma_3$, a vertex integral $\int_{V_i} dz \wedge dw$, and partial derivatives $\partial_z$, $\partial_w$ associated with the vertex coordinate that we integrate over.

The partial derivatives act on the two of the propagators emitting from the vertex $V_3$ to other vertices $V_2$, $V_4$, and effectively produce a multiplicative factor

$$\frac{\bar{z}_{32}\bar{w}_{34}}{d_{32}^2 d_{34}^2}. \tag{114}$$

Similarly, the partial derivatives that act on the propagator emitting from the vertex $V_4$ to a vertex $V_2$ and the external leg, and effectively produce a multiplicative factor

$$\frac{\bar{z}_{24}}{d_{24}^2}\partial_{w_4}, \tag{115}$$

where $\partial_{w_4}$ is assumed to act on $A_{ext}$.

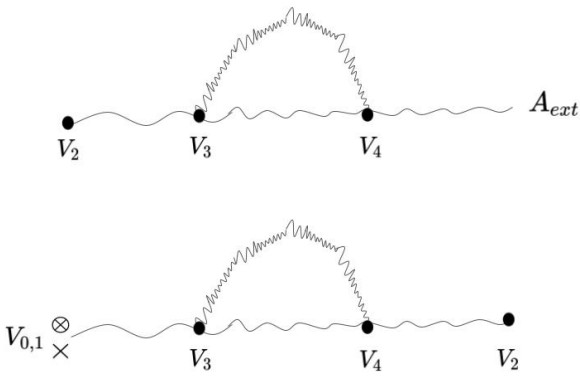

Figure 13: The first diagram is a loop correction on the external leg($\sim A_{ext}$) and the second diagram is a loop correction on one of the internal propagators $P_{02}$, $P_{12}$ that we have worked with in the previous subsections.

Considering the three new propagators $P_{32}P_{24}P_{43}$, the multiplicative factor introduced by the addition of the new bridge is

$$\sigma_3 \frac{\bar{z}_{32}\bar{w}_{34}\bar{z}_{24}}{d_{32}^2 d_{34}^2 d_{24}^2} P_{32} \wedge P_{24} \wedge P_{43} \partial_w. \tag{116}$$

The numerator of $P_{ij}$ is an anti-holomorphic 2-form on the 2-point configuration space of $V_i$ and $V_j$.

When we multiply the three propagators, recalling the definition (38), we see it is precisely zero.[16]

$$P_{32} \wedge P_{24} \wedge P_{43} = 0. \tag{117}$$

Since this vanishing property only depends on the three encircling propagators, the argument remains the same for the other two diagrams in Figure 12 that we have not discussed. Therefore, there is no higher loop correction on the internal vertex $V_2$.

Next, we consider the potential higher loop corrections on the external leg and the propagators. A similar analysis that was applied on Figure 12 goes through, and we see the multiplicative factor introduced by the new bridge is proportional to

$$P_{34} \wedge P_{43} \tag{118}$$

for both diagrams in Figure 13. Again, (118) is precisely zero. Therefore, there is no higher loop correction to the internal propagators and the external leg.

We conjecture that the vanishing phenomena[17] are generally the case for the combinations of anti-holomorphic 2-form of the 5d Chern-Simons propagators with their subscripts in the form of "a trace of a product of matrices ":

$$P_{i_1 i_2} \wedge P_{i_2 i_3} \wedge \ldots \wedge P_{i_n i_1} = 0. \tag{119}$$

# 4 Conclusion and open questions

We have reproduced the coproducts associated with the operator algebras on the intersecting M2 and M5 brane configuration in the $\Omega$-deformed M-theory by doing the perturbative

---

[16]We used a *Mathematica* package "grassmann" developed by Matthew Headrick in this computation.

[17]The similar vanishing phenomena were observed in 4d Chern-Simons theory [5].

computation in the 5d holomorphic-topological Chern-Simons theory.

There are some directions that we found interesting to pursue as future research.

- We sketched the fusion of the transverse surface defects by computing the 1-loop Feynman diagram. It would be nice to explicitly work it out by specifying the $\mathcal{W}_\infty$ algebra mode number associated to the vertices on two surface defects, and check if we indeed find the line operator as a byproduct of the fusion.

- Consider a codimension 1 surface defect in the 5d Chern-Simons theory that extends over $\mathbb{C}_z \times \mathbb{C}_w$. What degrees of freedom live on the defect, and how do they interact with the 5d Chern-Simons theory? To answer this question in some particular Calabi-Yau replacing $\mathbb{C}_{\epsilon_1} \times \mathbb{C}_{\epsilon_2} \times \mathbb{C}_{\epsilon_3}$, we may need to start with some physical 5d gauge theory and a 4d domain wall[18] and ask how they would interact to figure out the coupling Lagrangian. Then, we may proceed by applying the 5d version of topological holomorphic twist[19] on the coupled system.

    If we have an answer to the above question, can we set up some Feynman diagram rules for the domain wall? If so, what would be the heterotic fusion between the domain wall and the line defect? Ultimately, we would like to understand the Koszul duality for the domain wall.

- As we noticed in §2.3, $\rho : \mathcal{A} \to \mathcal{W}_\infty$ induces a Koszul dual embedding $^!\rho$ that maps a 5d gauge mode that couples to $\mathcal{W}_\infty$ into a 5d gauge mode that couples to $\mathcal{A}$. It would be interesting to find $^!\rho$ and its implications on the line and surface defect enriched 5d Chern-Simons theory.

# Acknowledgements

We thank Mathew Bullimore and another anonymous SciPost referee, who carefully read our draft and made many useful suggestions, which significantly improved the clarity of our paper. We also thank Miroslav Rapčák and Kevin Costello for comments on the first version of our paper.

**Funding information** Research of JO was supported by ERC Consolidator Grant number 864828 "Algebraic Foundations of Supersymmetric Quantum Field Theory (SCFTAlg)". Research at the Perimeter Institute is supported by the Government of Canada through Industry Canada and by the Province of Ontario through the Ministry of Economic Development & Innovation.

# A  Vertex coalgebra structure of $\mathcal{A}$

A vertex coalgebra consists of a vector space $V$ together with linear maps

- Coproduct $\lambda(w) : V \to (V \otimes V)((w^{-1}))$,

---

[18]Interesting examples of 4d $\mathcal{N} = 1$ domain walls in some 5d $\mathcal{N} = 1$ gauge theories were proposed in [57] with explicit superpotentials that couple the 4d theories on the domain walls and the 5d theories. However, it seems hard to realize those Neumann interfaces in the twisted M-theory set-up, since the candidate brane for the domain wall can not be an NS5 brane.

[19] [58] introduced a 4d version, focusing on its application in 4d $\mathcal{N} = 2$ version of geometric Langlands [59]. [60, 61] described the 3d version, concerning an algebraic structure under the theory, known as Poisson Vertex Algebra.

- Covacuum $c : V \to \mathbb{C}$,

satisfying the following axioms:

(1) Left counit: $\forall v \in V$,

$$(c \otimes \mathrm{Id}_V) \Lambda(w) v = v.$$

(2) Cocreation: $\forall v \in V$,

$$(\mathrm{Id}_V \otimes c) \Lambda(w) v \in V[w], \text{ and } \lim_{w \to 0} (\mathrm{Id}_V \otimes c) \Lambda(w) v = 0.$$

(3) Jacobi identity: $\forall v \in V$,

$$w_0^{-1} \delta \left( \frac{w_1 - w_2}{w_0} \right) (\mathrm{Id}_V \otimes \Lambda(w_2)) \Lambda(w_1) - w_0^{-1} \delta \left( \frac{w_2 - w_1}{-w_0} \right) (T \otimes \mathrm{Id}_V)$$

$$(\mathrm{Id}_V \otimes \Lambda(w_1)) \Lambda(w_2) = w_2^{-1} \delta \left( \frac{w_1 - w_0}{w_2} \right) (\Lambda(w_0) \otimes \mathrm{Id}_V) \Lambda(w_2),$$

where the formal delta series is

$$\delta \left( \frac{y - x}{z} \right) = \sum_{m \geq 0, n \in \mathbb{Z}} \binom{n}{m} y^{n-m} x^m z^{-n},$$

and

$$T : V_1 \otimes V_2 \to V_2 \otimes V_1 \tag{120}$$

is a linear map that swaps two components.

**Theorem 1.** *The M2 brane algebra $\mathcal{A}$ has a natural vertex coalgebra structure where the coproduct $\Lambda(w)$ is $\Delta_{\mathcal{A}} : \mathcal{A} \to (\mathcal{A} \otimes \mathcal{A})((w^{-1}))$ and the covacuum is the augmentation map $c : \mathcal{A} \to \mathbb{C}$ that sends $1$ to $1$ and $t_{a,b}$ to $0$ for all pairs $(a, b)$.*

Before proving the theorem, let us recall that $\Delta_{\mathcal{A}}$ acts on generators of $\mathcal{A}$ as [11]:

$$\Delta_{\mathcal{A}}(t_{0,n}) = 1 \otimes t_{0,n} + \sum_{m=0}^{n} \binom{n}{m} w^{n-m} t_{0,m} \otimes 1, \tag{121}$$

$$\Delta_{\mathcal{A}}(t_{2,0}) = 1 \otimes t_{2,0} + t_{2,0} \otimes 1 + 2\sigma_3 \sum_{m,n \geq 0} \frac{(m+n+1)!}{m!n!} (-1)^n w^{-n-m-2} t_{0,n} \otimes t_{0,m}. \tag{122}$$

By (11), we see that $\mathcal{A}$ is generated by $t_{2,0}$ and $t_{0,n}$. Since $\Delta_{\mathcal{A}}$ is an algebra homomorphism, the above formulae determine $\Delta_{\mathcal{A}}$ completely. Observe that the following three maps are algebra homomorphisms:

$$c \otimes \mathrm{Id} : (\mathcal{A} \otimes \mathcal{A})((w^{-1})) \to \mathcal{A}((w^{-1})), \tag{123}$$

$$\mathrm{Id} \otimes c : (\mathcal{A} \otimes \mathcal{A})((w^{-1})) \to \mathcal{A}((w^{-1})), \tag{124}$$

$$\lim_{w \to 0} : \mathcal{A}[w] \to \mathcal{A}. \tag{125}$$

The first two equations follow from the fact that $c : \mathcal{A} \to \mathbb{C}$ is an algebra homomorphism, and the third one is obvious. We will use these facts intensively in the proof of the theorem. Let us prove the three axioms in the following subsections:

## A.1 Left counit

Since $(c \otimes \mathrm{Id}_{\mathcal{A}}) \lambda(w) : \mathcal{A} \to \mathcal{A}((w^{-1}))$ is an algebra homomorphism, it suffices to show that $c$ is left counit when $v = t_{2,0}$ and $v = t_{0,n}$, which is done by direct computation:

$$(c \otimes \mathrm{Id}_{\mathcal{A}}) \lambda(w) t_{0,n} = (c \otimes \mathrm{Id}_{\mathcal{A}}) \left( 1 \otimes t_{0,n} + \sum_{m=0}^{n} \binom{n}{m} w^{n-m} t_{0,m} \otimes 1 \right) = t_{0,n}, \qquad (126)$$

and

$$(c \otimes \mathrm{Id}_{\mathcal{A}}) \lambda(w) t_{2,0} = (c \otimes \mathrm{Id}_{\mathcal{A}})(1 \otimes t_{2,0} + t_{2,0} \otimes 1) \qquad (127)$$

$$+ 2\sigma_3 \sum_{m,n \geq 0} \frac{(m+n+1)!}{m! n!} (-1)^n w^{-n-m-2} (c \otimes \mathrm{Id}_{\mathcal{A}}) \left( t_{0,n} \otimes t_{0,m} \right)$$

$$= t_{2,0}. \qquad (128)$$

Thus, we see that $(c \otimes \mathrm{Id}_{\mathcal{A}}) \lambda(w) v = v$, for all $v \in \mathcal{A}$.

## A.2 Cocreation

Again, it is enough to show that $(\mathrm{Id}_{\mathcal{A}} \otimes c) \lambda(w) t_{2,0}$ and $(\mathrm{Id}_{\mathcal{A}} \otimes c) \lambda(w) t_{0,n}$ are elements of $\mathcal{A}[w]$, and after taking $w \to 0$ limit they become $t_{2,0}$ and $t_{0,m}$ respectively. This is done by direct computation as well:

$$(\mathrm{Id}_{\mathcal{A}} \otimes c) \lambda(w) t_{0,n} = (\mathrm{Id}_{\mathcal{A}} \otimes c) \left( 1 \otimes t_{0,n} + \sum_{m=0}^{n} \binom{n}{m} w^{n-m} t_{0,m} \otimes 1 \right)$$

$$= \sum_{m=0}^{n} \binom{n}{m} w^{n-m} t_{0,m} \in \mathcal{A}[w]. \qquad (129)$$

Therefore,

$$\lim_{w \to 0} (\mathrm{Id}_{\mathcal{A}} \otimes c) \lambda(w) t_{0,n} = \lim_{w \to 0} \sum_{m=0}^{n} \binom{n}{m} w^{n-m} t_{0,m} = t_{0,n}. \qquad (130)$$

Furthermore,

$$(\mathrm{Id}_{\mathcal{A}} \otimes c) \lambda(w) t_{2,0} = (\mathrm{Id}_{\mathcal{A}} \otimes c)(1 \otimes t_{2,0} + t_{2,0} \otimes 1)$$

$$+ 2\sigma_3 \sum_{m,n \geq 0} \frac{(m+n+1)!}{m! n!} (-1)^n w^{-n-m-2} (\mathrm{Id}_{\mathcal{A}} \otimes c) \left( t_{0,n} \otimes t_{0,m} \right)$$

$$= t_{2,0}, \qquad (131)$$

and the $w \to 0$ limit is of course $t_{2,0}$. Hence, we see that $(\mathcal{A}, \Delta_{\mathcal{A}}, c)$ satisfies the cocreation axiom.

## A.3 Jacobi identity

This is the hardest part. We will use an equivalent form of Jacobi identity proven in [62]. Let us first record the relevant result (Proposition 3.2 of [62]):

- In the presence of all axioms of a graded vertex coalgebra except the Jacobi identity, weak cocommutativity and the $D^*$-bracket identity imply the Jacobi identity.

For now, let us omit the term "graded" vertex coalgebra and only focus on the weak cocommutativity and the $D^*$-bracket. We will come back to the grading at the end. We will prove that $\Delta_{\mathcal{A}}$ satisfies the cocommutativity which implies the weak cocommutativity.

We will first show the $D^*$-bracket identity. By Proposition 2.2 of [62], the linear operator $D^* : \mathcal{A} \to \mathcal{A}$ is defined as

$$D^* = \mathrm{Res}_w w^{-2}(\mathrm{Id}_{\mathcal{A}} \otimes c) \lambda(w). \tag{132}$$

In other words, by applying $D^*$ on $v$, we pick up the linear term of $(\mathrm{Id}_{\mathcal{A}} \otimes c) \lambda(w) v \in \mathcal{A}[w]$. On the generators of $\mathcal{A}$, $D^*$ acts as follows:

$$D^*(t_{0,n}) = n \cdot t_{0,n-1} \quad \text{and} \quad D^*(t_{2,0}) = 0. \tag{133}$$

Notice that $D^*$ acts on $\mathcal{A}$ as a derivation:

$$
\begin{aligned}
D^*(v_1 v_2) &= \mathrm{Res}_w w^{-2} \left(v_1 + D^*(v_1)w + \mathcal{O}(w^2)\right)\left(v_2 + D^*(v_2)w + \mathcal{O}(w^2)\right) \\
&= D^*(v_1)v_2 + v_1 D^*(v_2). \tag{134}
\end{aligned}
$$

Thus, the action of $D^*$ on $t_{2,0}$ and $t_{0,n}$ completely determines its action on $\mathcal{A}$. It is not hard to see that

$$D^*(t_{a,b}) = b \cdot t_{a,b-1}. \tag{135}$$

The $D^*$-bracket identity (Equation (2.20) of [62]) that we are going to show is defined as

$$\frac{\mathrm{d}}{\mathrm{d}w} \lambda(w) = \lambda(w)D^* - (\mathrm{Id}_{\mathcal{A}} \otimes D^*)\lambda(w). \tag{136}$$

Observe that both the left hand side and the right hand side of the above equation, when treated as maps between $\mathcal{A}$ and $(\mathcal{A} \otimes \mathcal{A})((w^{-1}))$, are derivations. Thus, we only need to show that the $D^*$-bracket identity holds for generators $t_{0,n}$ and $t_{2,0}$. This is achieved by direct computation:

$$\lambda(w)D^*(t_{0,n}) - (\mathrm{Id}_{\mathcal{A}} \otimes D^*)\lambda(w)t_{0,n} \tag{137}$$

$$= n\lambda(w)t_{0,n-1} - (\mathrm{Id}_{\mathcal{A}} \otimes D^*)\left(1 \otimes t_{0,n} + \sum_{m=0}^{n} \binom{n}{m} w^{n-m} t_{0,m} \otimes 1\right)$$

$$= n\left(1 \otimes t_{0,n-1} + \sum_{m=0}^{n-1} \binom{n-1}{m} w^{n-m-1} t_{0,m} \otimes 1\right) - n \cdot 1 \otimes t_{0,n-1}$$

$$= \sum_{m=0}^{n-1} \frac{n!}{(n-m-1)!m!} w^{n-m-1} t_{0,m} \otimes 1$$

$$= \frac{\mathrm{d}}{\mathrm{d}w} \lambda(w)t_{0,n}, \tag{138}$$

and

$$\lambda(w)D^*(t_{2,0}) - (\mathrm{Id}_{\mathcal{A}} \otimes D^*)\lambda(w)t_{2,0} \tag{139}$$

$$= -(\mathrm{Id}_{\mathcal{A}} \otimes D^*)\left(2\sigma_3 \sum_{m,n \geq 0} \frac{(m+n+1)!}{m!n!} (-1)^n w^{-n-m-2} t_{0,n} \otimes t_{0,m}\right)$$

$$= -2\sigma_3 \sum_{m \geq 1, n \geq 0} \frac{(m+n+1)!}{(m-1)!n!} (-1)^n w^{-n-m-2} t_{0,n} \otimes t_{0,m-1} \tag{140}$$

$$= -2\sigma_3 \sum_{m,n \geq 0} \frac{(m+n+2)!}{(m!n!} (-1)^n w^{-n-m-3} t_{0,n} \otimes t_{0,m} \tag{141}$$

$$= \frac{\mathrm{d}}{\mathrm{d}w} \lambda(w)t_{2,0}. \tag{142}$$

Hence, we see that $\mathcal{A}$ satisfies the $D^*$-bracket identity.

Next, we explain what cocommutativity is. Let us introduce some notations first. Let

$$\iota_{12} : \mathbb{C}[[x_1^{-1}, x_2^{-1}, (x_1 - x_2)^{-1}]][x_1, x_2] \hookrightarrow \mathbb{C}[[x_1, x_2, x_1^{-1}, x_2^{-1}]] \tag{143}$$

be a map that sends $\frac{f(x_1,x_2)}{x_1^r x_2^s (x_1-x_2)^t}$ to $\frac{f(x_1,x_2)}{x_1^r x_2^s}$ times $\frac{1}{(x_1-x_2)^t}$ expanded in non-negative powers of $x_2$, where $f(x_1, x_2) \in \mathbb{C}[x_1, x_2]$ and $r, s, t \in \mathbb{N}$. Then the cocommutativity can be stated as follows:

$$\forall v \in \mathcal{A}, \ (\mathrm{Id}_{\mathcal{A}} \otimes \Lambda(w_2))\Lambda(w_1)v \ \text{ and } \ (T \otimes \mathrm{Id}_{\mathcal{A}})(\mathrm{Id}_{\mathcal{A}} \otimes \Lambda(w_1))\Lambda(w_2)v \in \mathrm{Im}(\iota_{21}). \tag{144}$$

Moreover,

$$\iota_{12}^{-1}(\mathrm{Id}_{\mathcal{A}} \otimes \Lambda(w_2))\Lambda(w_1)v = \iota_{21}^{-1}(T \otimes \mathrm{Id}_{\mathcal{A}})(\mathrm{Id}_{\mathcal{A}} \otimes \Lambda(w_1))\Lambda(w_2)v. \tag{145}$$

Since both $(\mathrm{Id}_{\mathcal{A}} \otimes \Lambda(w_2))\Lambda(w_1)$ and $(T \otimes \mathrm{Id}_{\mathcal{A}})(\mathrm{Id}_{\mathcal{A}} \otimes \Lambda(w_1))\Lambda(w_2)$ are algebra homomorphisms, we only need to prove the cocommutativity by showing that (145) holds for $t_{0,n}$ and $t_{2,0}$. Again, this is done by direct computations:

$$(\mathrm{Id}_{\mathcal{A}} \otimes \Lambda(w_2))\Lambda(w_1)t_{0,n} \tag{146}$$

$$= (\mathrm{Id}_{\mathcal{A}} \otimes \Lambda(w_2))\left( 1 \otimes t_{0,n} + \sum_{m=0}^{n} \binom{n}{m} w_1^{n-m} t_{0,m} \otimes 1 \right)$$

$$= 1 \otimes 1 \otimes t_{0,n} + \sum_{m=0}^{n} \binom{n}{m} w_1^{n-m} t_{0,m} \otimes 1 \otimes 1 + \sum_{m=0}^{n} \binom{n}{m} w_2^{n-m} 1 \otimes t_{0,m} \otimes 1, \tag{147}$$

which is apparently in the image of $\iota_{12}$. Swapping $w_1$ and $w_2$ and applying $T \otimes \mathrm{Id}_{\mathcal{A}}$, we see that

$$(T \otimes \mathrm{Id}_{\mathcal{A}})(\mathrm{Id}_{\mathcal{A}} \otimes \Lambda(w_1))\Lambda(w_2)t_{0,n}$$

$$= 1 \otimes 1 \otimes t_{0,n} + \sum_{m=0}^{n} \binom{n}{m} w_1^{n-m} t_{0,m} \otimes 1 \otimes 1 + \sum_{m=0}^{n} \binom{n}{m} w_2^{n-m} 1 \otimes t_{0,m} \otimes 1, \tag{148}$$

which agrees with $(\mathrm{Id}_{\mathcal{A}} \otimes \Lambda(w_2))\Lambda(w_1)t_{0,n}$.

$$(\mathrm{Id}_{\mathcal{A}} \otimes \Lambda(w_2))\Lambda(w_1)t_{2,0}$$

$$= (\mathrm{Id}_{\mathcal{A}} \otimes \Lambda(w_2))\left( 1 \otimes t_{2,0} + t_{2,0} \otimes 1 + 2\sigma_3 \sum_{m,n \geq 0} \frac{(m+n+1)!}{m!n!}(-1)^n w_1^{-n-m-2} t_{0,n} \otimes t_{0,m} \right)$$

$$= t_{2,0} \otimes 1 \otimes 1 + 1 \otimes 1 \otimes t_{2,0} + 1 \otimes t_{2,0} \otimes 1$$

$$+ 2\sigma_3 \sum_{m,n \geq 0} \frac{(m+n+1)!}{m!n!}(-1)^n w_2^{-n-m-2} 1 \otimes t_{0,n} \otimes t_{0,m}$$

$$+ 2\sigma_3 \sum_{m,n \geq 0} \frac{(m+n+1)!}{m!n!}(-1)^n w_1^{-n-m-2} t_{0,n} \otimes 1 \otimes t_{0,m}$$

$$+ 2\sigma_3 \sum_{m,n \geq 0} \sum_{k=0}^{m} \frac{(m+n+1)!}{m!n!}(-1)^n \binom{m}{k} w_1^{-n-m-2} w_2^{m-k} t_{0,n} \otimes t_{0,k} \otimes 1. \tag{149}$$

Let us focus on the last line. Fix $n$ and $k$, let $a = m - k$, and do the summation for $a$ running

from 0 to $\infty$:

$$\sum_{a\geq 0} \frac{(a+n+k+1)!}{(a+k)!n!}(-1)^n \binom{a+k}{k} w_1^{-n-k-2} \left(\frac{w_2}{w_1}\right)^a$$

$$= \frac{(n+k+1)!}{k!n!}(-1)^n w_1^{-n-k-2} \sum_{a\geq 0} \frac{(a+n+k+1)!}{a!} \left(\frac{w_2}{w_1}\right)^a$$

$$= \frac{(n+k+1)!}{k!n!}(-1)^n w_1^{-n-k-2} \left(\frac{1}{1-\frac{w_2}{w_1}}\right)^{n+k+2}$$

$$= \frac{(n+k+1)!}{k!n!}(-1)^n \left(\frac{1}{w_1-w_2}\right)^{n+k+2}. \tag{150}$$

Therefore, the last line of (149) is

$$2\sigma_3 \sum_{n,k\geq 0} \frac{(n+k+1)!}{k!n!}(-1)^n \left(\frac{1}{w_1-w_2}\right)^{n+k+2} t_{0,n} \otimes t_{0,k} \otimes 1. \tag{151}$$

Thus, we see that $(\mathrm{Id}_{\mathcal{A}} \otimes \lambda(w_2))\lambda(w_1)t_{2,0}$ is in the image of $\iota_{12}$. Swapping $w_1$ and $w_2$ and applying $T \otimes \mathrm{Id}_{\mathcal{A}}$, we see that

$$(T \otimes \mathrm{Id}_{\mathcal{A}})(\mathrm{Id}_{\mathcal{A}} \otimes \lambda(w_1))\lambda(w_2)t_{2,0}$$

$$= t_{2,0} \otimes 1 \otimes 1 + 1 \otimes 1 \otimes t_{2,0} + 1 \otimes t_{2,0} \otimes 1$$

$$+ 2\sigma_3 \sum_{m,n\geq 0} \frac{(m+n+1)!}{m!n!}(-1)^n w_2^{-n-m-2} 1 \otimes t_{0,n} \otimes t_{0,m}$$

$$+ 2\sigma_3 \sum_{m,n\geq 0} \frac{(m+n+1)!}{m!n!}(-1)^n w_1^{-n-m-2} t_{0,n} \otimes 1 \otimes t_{0,m}$$

$$+ 2\sigma_3 \sum_{n,k\geq 0} \frac{(n+k+1)!}{k!n!}(-1)^n \left(\frac{1}{w_2-w_1}\right)^{n+k+2} t_{0,k} \otimes t_{0,n} \otimes 1, \tag{152}$$

which agrees with $(\mathrm{Id}_{\mathcal{A}} \otimes \lambda(w_2))\lambda(w_1)t_{2,0}$. This concludes the proof that $\mathcal{A}$ satisfies the co-commutativity.

Finally, to apply Proposition 3.2 of [62], we need that $\mathcal{A}$ is graded in the sense of Proposition 2.3 of [62]. In particular, $\mathcal{A}$ must be written as $\mathcal{A} = \coprod_{k\geq -N} \mathcal{A}_{(k)}$ and for $v \in \mathcal{A}_{(r)}$ we have

$$\lambda(w)v = \sum_{k\in\mathbb{Z}} \Delta_k(v)w^{-k-1}, \tag{153}$$

where $\Delta_k(v) \in (\mathcal{A} \otimes \mathcal{A})_{r+k+1}$ for each $k \in \mathbb{Z}$. Tentatively, let us grade $\mathcal{A}$ by setting the degree of the generators as

$$\deg(1) = 0, \ \deg(t_{a,b}) = b - a. \tag{154}$$

Then the coproduct of $\mathcal{A}$ satisfies (153). This grading is not bounded below; however, the definition requires boundedness. Nevertheless, we can resolve this issue by generalizing the definition of graded vertex coalgebra as follows.

We say that a vertex coalgebra $V$ is graded if $V = \coprod_{k\in Z} V_{(k)}$ and (153) is satisfied. Moreover, there is a universal constant $M$ such that

$$\Delta_k(v) \in (\oplus_{j\geq\min\{M,r\}} V_{(j)}) \otimes (\oplus_{j\geq\min\{M,r\}} V_{(j)}), \tag{155}$$

for all $k$. Note that the old definition in [62] is contained in the new definition, since we can set $M = N$.

We claim that by relaxing the definition of the graded vertex coalgebra to ours, all results in sections 2 and 3 of [62] still hold. The reason is that the only place where the boundedness assumption of the old definition is used is to show the boundedness of the homogeneous components of $(\mathrm{Id}_V \otimes \lambda(w_1)) \lambda(w_2) v$ and $(\lambda(w_1) \otimes \mathrm{Id}_V) \lambda(w_2) v$. Instead of a hard cut-off by setting $V_{(k)} = 0$ for $k \ll 0$, we can do it by requiring components of $\lambda(w) v$ bounded below and hence components of $(\mathrm{Id}_V \otimes \lambda(w_1)) \lambda(w_2) v$ and $(\lambda(w_1) \otimes \mathrm{Id}_V) \lambda(w_2) v$ bounded below.

Now $\mathcal{A}$ becomes graded, since we can set $M = 0$ and observe that

$$\Delta_k(t_{2,0}) \in (\oplus_{j \geq -2} \mathcal{A}_{(j)}) \otimes (\oplus_{j \geq -2} \mathcal{A}_{(j)}),$$
$$\Delta_k(t_{0,n}) \in (\oplus_{j \geq 0} \mathcal{A}_{(j)}) \otimes (\oplus_{j \geq 0} \mathcal{A}_{(j)}),$$

which means that (155) is satisfied for $v = t_{2,0}, t_{0,n}$. It is straightforward to see that if $v_1, v_2$ satisfy (155), then $v_1 v_2$ also satisfies (155). Thus, $\mathcal{A}$ satisfies (155). We conclude the proof of the Jacobi identity by applying Proposition 3.2 of [62], modified with the new definition of the graded vertex coalgebra.

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
