# Peer review of "Twisted holography of defect fusions"

_SciPost Physics, doi:SciPost Phys. 10, 105 (2021)_

## Round 1 · Referee Report · Anonymous (Referee 1) · 2021-3-30

Strengths

The paper fills a gap in the recent twisted holography literature by performing a bulk version of computations of defect fusions (mathematically, coproducts) performed in [11] using purely algebraic means.

1.) The computations of coproducts are presented in detail and successfully reproduce the main results of [11].

2.) The paper proposes (the beginning of) a generalization to the fusion relations of [11] to the case where two surface defects are oriented transversely, and sketch a computation that verifies the expectation that the fusion will produce a line defect.

Weaknesses

1.) The paper is not easily digestible to readers not familiar with the prior literature on twisted holography in the setting of Omega-deformed M-theory.

Report

I would recommend this paper after some physical points are clarified and various minor typos are corrected.

Requested changes

1.) At the beginning of section 2, the authors describe a "bosonic ghost with a vanishing nilpotent vev", though a vev is a scalar quantity and therefore declaring it as nilpotent is confusing. The authors should clarify this point.

2.) At the bottom of page 4 the authors discuss large N, N' limits used in taking the holographic duality. Are both N, N' required to be large simultaneously for this duality, or can one take either N or N' large (so that, e.g., large numbers of M2 branes source the 'boundary' theory and one can study a finite number N' of defects added to the dual pair)?

3.) At the top of page 5, the authors refer to "the operator algebra of the field theory" arising from Koszul duality. Do they more properly mean part of the operator algebra of the twisted field theory, or is the claim that Koszul duality determines all OPEs of the twisted field theory? If the latter, the authors should explain why this is true.

4.) On page 5 the authors describe the gauge algebra of the 5d bulk Chern-Simons theory that is the remnant of twisted M-theory, and this gauge algebra depends on N', the number of M5 branes. A priori, the N' M5 branes are some additional defects inserted into the system, so the authors should explain in more detail how the bulk algebra arises and why it is isomorphic to the claimed algebra.

5.) In section 2.1, the authors mention that the Omega background quantizes the chiral ring, and should clarify that it is a deformation, rather than a geometric, quantization.

6.) In section 2.1, the authors describe the algebra supported on the twisted, Omega-deformed M2 brane. Since the M2 brane is supported on \epsilon 1, the appearances of \epsilon2 and 3 in various places is not obvious. From the point of view of the M2 brane algebra, \epsilon 2, 3 naively look like they should appear symmetrically, since they are both transverse to the M2 brane, but they appear quite distinct. The authors should explain the appearances of \epsilon 2, 3 in formulas 9 (where it looks like the TQM is already coupled to the 5d CS theory), 11, and in the paragraphs after equation 12.

7.) Below equation 13 the authors write the BRST transformation as A-> c, and it would be preferable to spell out the precise BRST transformation (and why, when determining which diagrams to study, it suffices to only replace A with c as indicated).

8.) The logic behind section 2.3 is a bit unclear and should be spelled out more. Naively, one can start with two versions of Koszul duality for the bulk 5d CS algebra (a Koszul dual associative algebra in a topological direction along the line, and a Koszul dual vertex algebra in holomorphic directions along a surface). Is the claim in 2.3 that since one can make an associative algebra from the W\infty mode algebra, there must be an isomorphism to the associative algebra A? What are the physical and algebraic reasons the morphisms (17) must exist?

9.) Just above equation 19, the authors mention that Koszul duality induces a coproduct structure, which is explained mathematically in the footnote. There should also be a simple physical explanation from the bulk point of view that a coproduct exists (Feynman diagrams with bulk lines connecting to the two defects participating in the fusion) that should be described.

10.) There are numerous minor typos. It would be too much to make an exhaustive list, but for instance:

a.) In the second paragraph, 'as forming' should be 'to form'.
b.) In the third paragraph 'philosophy under our approach' should be 'philosophy of our approach'
c.) 'a gauge invariance' should be 'gauge invariance' in the paragraph preceding the Plan of the Paper
d.) In numerous places Obs is written as ObS (unless this is intentional?)
e.) In 'The homogeneous fusion' section, the authors should write "a fusion of line defects *with each other*" (similarly with surface defects) for clarity.

---

## Round 1 · Referee Report · Mathew Bullimore (Referee 2) · 2021-4-27

Strengths

1) The paper successfully reproduces a proposal in reference [11] for the co-products arising from the fusion of M2 brane and M5 brane defects in the setup of twisted holography for omega deformed M-theory.

Weaknesses

1) Section 2 provided a summary of the necessary background for the computations in the paper, but as someone not intimately familiar with references [1-11], there were a number of points that would benefit from some additional explanations. The comments below are mainly aimed at improving this aspect of the paper.

Report

I recommend the paper to be published subject to minor changes.

Requested changes

1) In the paragraph around equation (6), the origin and role of the parameter N' was not obvious to me. Is it an auxiliary parameter or the same N' giving the number of M5 branes? If the latter, why is it relevant for the M2 brane algebra A? Why is gl(N') Chern-Simons appearing here?

2) At the end of section 2.1, the precise relation between A and the enveloping algebra of fluctuations is not clear. The second to last paragraph on page 6 suggests they are equal. If so, does this follow from the coupling in equation (13)?

3) The paper frequently refers to a BRST transformation $A\to c$. Could the authors clarify what is meant by this? A BRST transformation might usually be of the form $\delta A = d c$.

4) Could the author's clarify what is the relationship between the numerical coefficients $c_{m,n}$ and $d_{m,n}$?

5) The paper made clear the exact values of the coefficients $d_{m,n}$ was not important - but it wasn't clear to me whether the holographic computation exactly reproduced these coefficients computed via the algebraic method or not. A clear statement would help.

6) Section 2.3 introduced an embedding map $\rho$. What is the physical picture behind this?

7) Here are a couple of typos that I came accross: i) At the top of page 5: "holomoprhically". ii) In footnote 3, should it refer to equation (3) instead of (1)? iii) The sentence at the top of page 5 doesn't quite make sense to me. iv) At the start of section 3.1: " we approach them together" should be something like "when they approach each other".

---

## Round 2 · Referee Report · Anonymous (Referee 1) · 2021-5-5

Report

The authors incorporated many suggestions from the previous version of the report and improved the clarity of their draft. I am happy to recommend this version for publication.

---

## Round 2 · Referee Report · Anonymous (Referee 3) · 2021-5-7

Report

I am happy to recommend the updated version for publication.

---

## Round 2 · Author Response

Dear SciPost referees,

We are very grateful to your careful reading of our paper and thoughtful comments.

We have tried our best to address all points that were asked to clarify.

Best,
Jihwan

---

## Editorial Decision

published